# Human brain dynamics and spatiotemporal trajectories during threat processing

Joyneel Misra[1]*, Luiz Pessoa[1,2,3]*

[1]Department of Electrical and Computer Engineering, University of Maryland, College Park, United States; [2]Department of Psychology, University of Maryland, College Park, United States; [3]Maryland Neuroimaging Center, University of Maryland, College Park, United States

## eLife Assessment

Using highly sophisticated switching linear dynamical systems (SLDS) analyses applied to functional MRI data, this study provides **important** insights into network dynamics underlying threat processing. After identifying distinct neural network states associated with varying levels of threat proximity, the paper provides **compelling** evidence of intrinsically and extrinsically driven contributions to these within-state dynamics and between-state transitions. Although the findings could be made more biologically meaningful, this work will be of interest to a wider functional neuroimaging and systems neuroscience community.

**\*For correspondence:**
joyneelm@umd.edu (JM);
pessoa@umd.edu (LP)

**Competing interest:** The authors declare that no competing interests exist.

**Abstract** Functional MRI (fMRI) research has traditionally investigated task processing using static blocked or event-related designs. Consequently, our understanding of threat processing remains limited to findings from paradigms with restricted dynamics. In this paper, we applied switching linear dynamical systems (SLDSs) to uncover the dynamics of threat processing during a continuous threat-of-shock paradigm. Unlike typical systems neuroscience studies that assume systems are decoupled from external inputs, we characterized both endogenous and exogenous contributions to the dynamics. We first demonstrated that the SLDS model learned the regularities of the experimental paradigm; states and state transitions estimated from fMRI data across 85 regions of interest reflected both threat proximity and direction (approach vs. retreat). After establishing that the model captured key properties of threat-related processing, we characterized the dynamics of states and their transitions. The results reveal how threat processing can be viewed as dynamic multivariate patterns whose trajectories are determined by intrinsic and extrinsic factors that jointly drive how the brain temporally evolves. Furthermore, we developed a measure of region importance to quantify individual brain region contributions to system dynamics, complementing the system-level SLDS formalism. Finally, we demonstrated that an SLDS model trained on one paradigm successfully generalizes to a separate experiment, capturing fMRI dynamics across distinct threat-processing tasks. We propose that viewing threat processing through the lens of dynamical systems offers vital avenues to uncover properties of threat dynamics not unveiled by standard experimental designs.

## Introduction

Historically, systems neuroscience has sought to develop experimental paradigms where an effect of interest is varied while as many other variables remain fixed in an attempt to isolate the impact of the former while controlling for the latter. Such a time-tested strategy has provided deep insights

into brain and behavior. However, with the advent of neurotechnologies allowing the study of animals in more freely behaving conditions, more naturalistic paradigms have been introduced (*Hunt et al., 2021*; *Testard et al., 2024*). At the same time, in human neuroimaging, researchers have defended the idea that dynamic and/or naturalistic paradigms are needed to advance our understanding of brain and behavior in a way that is not entirely constrained by rigid paradigms (*Hasson and Honey, 2012*; *Sonkusare et al., 2019*; *Finn et al., 2020*; *Lee Masson and Isik, 2021*; *Grall and Finn, 2022*). While such ideas resonate with those from ecological psychology and ethological approaches (*Gothard et al., 2018*; *Mobbs et al., 2018*; *Evans et al., 2019*), they introduce considerable challenges in the analysis of brain data and associated behaviors. If there are no specific trials or blocks in an experiment, what should constitute the unit of analysis? If the paradigm is more open-ended and lacks specific conditions (e.g. 'attend to the visual stimulus' vs. 'attend to the audio stimulus'), how should brain and behavior be investigated?

A promising approach is the use of unsupervised methods that consider time series dynamics (*Anderson and Fincham, 2014*; *Vidaurre et al., 2016*; *Baldassano et al., 2017*; *Linderman et al., 2016*; *Taghia et al., 2017*; *Pandarinath et al., 2018*; *Townsend and Gong, 2018*; *Morioka et al., 2020*; *Wu et al., 2021*; *Singh et al., 2022*; *Song et al., 2023*; *Xu et al., 2023*; *Vahidi et al., 2024*; *Chen et al., 2024*). In particular, hidden Markov models (HMMs) can be applied to time series data and partition it into 'states' in time. For example, in one study, continuous narratives were presented and state transitions were identified from functional MRI (fMRI) signals (*Baldassano et al., 2017*). Notably, state boundaries corresponded to human-annotated ones better than chance, and the match between model and human boundaries increased for more 'high-level' brain regions (e.g. angular gyrus).

Another study attempted to determine mental states while participants solved novel mathematical problems (*Anderson and Fincham, 2014*). Model states were interpreted as reflecting 'encoding', 'planning', 'solving', and 'responding' stages based on fMRI data. Although the authors found evidence for the detected states, the study also highlights some of the challenges of applying unsupervised methods to complex paradigms. The estimated duration of the final responding state correlated well with the measured time participants spent entering their responses (*r*=0.53), providing some validation of that state. However, the mapping between brain signals and putative mental states (e.g. 'encoding') remained speculative. More generally, state-based modeling of fMRI data would benefit from evaluation in contexts where the experimental paradigm affords a clearer mapping between discovered states and experimental manipulation, as done here.

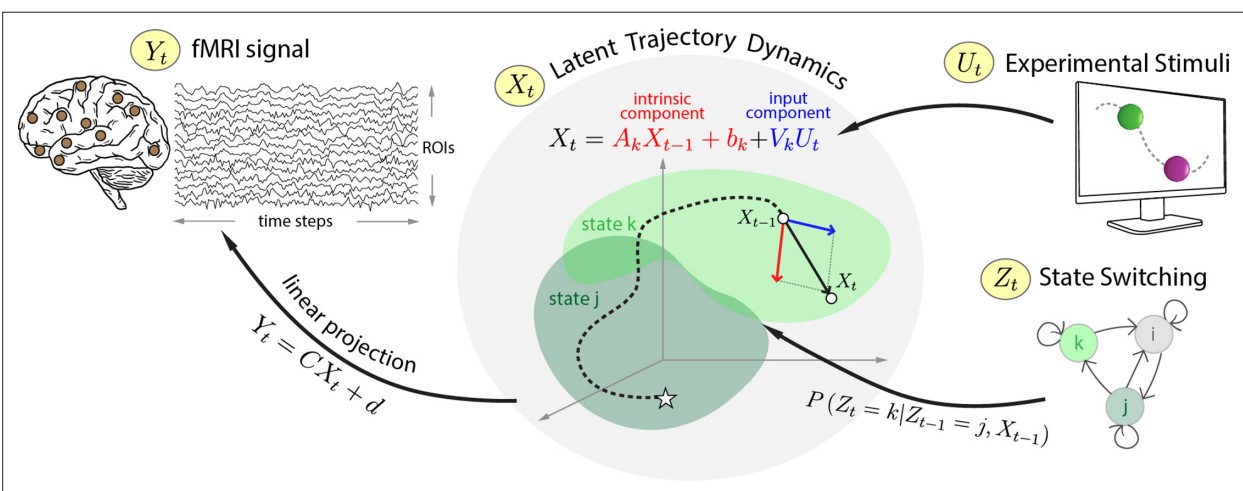

**Figure 1.** Determining shared dynamics during threat processing. Switching linear dynamical systems (SLDSs) were used to model functional MRI (fMRI) time series data ($Y_t$) from a set of brain regions of interest (ROIs). The framework assumes that time series data can be segmented into a set of discrete states. The model represents brain signals in terms of a set of latent variables ($X_t$). For each state $k$, the temporal evolution of the system is specified via a linear dynamical system with both intrinsic and input-related components. In the diagram, the system starts in state $j$ (white star) and transitions to state $k$ (the colored patches in the middle represent the subspaces associated with the two states). In state $k$, the system evolves according to the dynamics matrix $A_k$ and input contributions ($V_k$). Overall, as states switch temporally, so does the corresponding linear dynamical system governing the system's trajectory.

Overall, despite encouraging results (*Anderson and Fincham, 2014*; *Vidaurre et al., 2016*; *Baldassano et al., 2017*; *Taghia et al., 2017*; *Taghia et al., 2018*; *van de Meer et al., 2020*; *Wu et al., 2021*; *Song et al., 2021*; *Song et al., 2023*), the extent to which state-space models successfully parse fMRI data into meaningful mental/brain states remains poorly understood. Critically, while state-space models like HMMs are able to segment time series data, they do not enable discovery of properties of state dynamics. In particular, how do brain signals evolve during a brain state that is maintained for a period of time? In systems neuroscience, progress has been made in this context by studying brain signals via dynamical systems theory (*Yu et al., 2005*; *Deco et al., 2008*; *Shenoy et al., 2013*; *Remington et al., 2018*; *Robson and Li, 2022*). In particular, it has been suggested that attractor dynamics plays important computational functions (*Miller, 2016*; *Khona and Fiete, 2022*). However, comparable progress in fMRI research has lagged behind, perhaps because of the slowly evolving nature of the hemodynamic response (but see *John et al., 2022*).

Threat processing is typically studied in terms of discrete events and/or blocks that correspond to experimental conditions (but see *Mobbs et al., 2007*; *Mobbs et al., 2009*; *Murty et al., 2023*; *Levitas and James, 2024*). For example, a conditioned threat is presented and associated evoked brain responses determined (*Fullana et al., 2016*). But in many real-world situations, conditions involving threat unfold temporally in a less-discrete manner (*Fanselow and Lester, 1987*). In the present investigation, we investigated brain dynamics during a continuous threat-of-shock paradigm (*Meyer et al., 2019*; *Limbachia et al., 2021*; *Figure 1*). We sought to characterize the evolution of brain signals in terms of trajectories in multidimensional space, a framework at times called 'computing with trajectories' (*Buonomano and Maass, 2009*). Instead of responses to discrete events possibly observed across multiple brain sites, we determined multivariate and distributed patterns of activity with shared dynamics.

We employed a switching linear dynamical systems (SLDSs) framework that estimates a generative model of the data with both states and transitions like HMMs (*Ackerson and Fu, 1970*; *Barber, 2006*; *Fox et al., 2008*; *Linderman et al., 2019*). In addition, each state is described in terms of a linear dynamical system that specifies the trajectory of brain activity during the state. Importantly, we estimated both endogenous and exogenous components of the dynamics, whereas some past work has not modeled both contributions (see discussion in *Khona and Fiete, 2022*). A considerable challenge in state-based modeling, including SLDSs, is linking estimated states and dynamics to interpretable processes. Here, we developed a measure of *region importance* that provides a biologically meaningful way to bridge this gap, as it quantifies how individual brain regions contribute to steering state trajectories. Our approach demonstrated that we can recover key properties of threat-related processing in the brain (*Grupe and Nitschke, 2013*), while establishing novel properties of threat and safety dynamics. More generally, the approach can be applied to a broad range of continuous and/or naturalistic fMRI paradigms investigated in humans.

We also investigated model generalizability: how well does a model estimated in one paradigm generalize to related experiments? Examining this question is of importance because even statistically significant fits do not necessarily generalize well to unseen data, not to mention to related but different paradigms. However, if an SLDS model captures meaningful spatiotemporal regularities associated with behavioral/brain states in a given dataset, the SLDS states should provide good fits to unseen data from a second, related experiment. Here, we investigated the extent to which the SLDS model that was fit to the moving circles paradigm generalized to a second, continuous paradigm involving threat and safety conditions.

Is the present contribution focused on threat processing or methodological developments for the analysis of more continuous/ecologically valid paradigms? Our answer is 'both'. One goal was to contribute to the development of a framework that considers brain processing to be inherently dynamic and multivariate. In particular, our goal was to provide the formal basis for conceptualizing threat processing as a dynamic process (see *Fanselow and Lester, 1987*) subject to endogenous and exogenous contributions. At the same time, our study revealed how regions studied individually in the past (e.g. anterior insula, cingulate cortex) contribute to brain states with multiregion dynamics.

# Results

## Continuous threat processing switches between states

Participants watched two circles of different colors on the screen, at times moving close to each other, at times moving apart in a smooth but unpredictable fashion (*Meyer et al., 2019*; *Limbachia et al., 2021*). Upon circle collision, the participants received mild but unpleasant electrical stimulation together with an aversive sound. To enhance unpredictability and hence anxious apprehension by participants, the movement of the circles included multiple instances of 'near misses' in which the circles nearly touched before retreating from each other.

Time series data were analyzed according to the SLDS framework (*Figure 1*). Briefly, an SLDS estimates a model of the data using a set of unobserved (also called latent) variables assumed to govern system dynamics. The latent space is typically of considerably smaller dimension relative to the original data space (here 10 latent dimensions and 85 region of interest [ROI] time series, respectively). SLDSs consist of one linear dynamical system model per state, such that the trajectory of the system is governed by a single dynamical system at a time. Like HMMs, an SLDS segments the data into a fixed number of states, such that the transitions between states follow a Markov process. At a given time, a single state is considered to be 'active'. Overall, SLDS models provide a piecewise linear approximation of (potentially nonlinear) dynamics. Time series data were obtained from 85 regions involved in threat-related processing, as defined in a prior study (*Murty et al., 2022*).

The results reported below were obtained with 92 participants. A separate group of 30 participants was used to define hyperparameters, including the number of states ($K = 6$, see Appendix 1) and the dimensionality of the latent space ($D = 10$, see Appendix 1). Assessing statistical significance when complex models are applied to multi-participant data requires estimating variability of the estimates across participants. We employed a bootstrap procedure such that an SLDS model was estimated for each bootstrap sample, allowing us to estimate how SLDS parameters varied across the population (see Methods), thus allowing generalization beyond the current sample.

## Model states are synchronous with the experimental paradigm

We evaluated if the states estimated by the model reflected experimental stimuli. For example, when the circles were far from each other, participants were 'safe', in contrast to when the circles were rather close and could progress to a collision. We encoded the input based on 20 bins defined in terms of circle proximity and direction: $A_1 - A_{10}$ for approach, $R_1 - R_{10}$ for retreat (see Methods, top inset in *Figure 2*). When the circles approached each other, $A_1$ indicated the farthest distance and $A_{10}$ indicated that they collided; when they retreated from each other, $R_{10}$ indicated that the circles collided and $R_1$ indicated the farthest distance. We determined the associations between brain states and stimulus categories (i.e. bins) by calculating the probability of a brain state given the stimulus, $P(\text{state} \mid \text{stimulus})$ (see Methods). The probabilities are shown in *Figure 2A* (e.g. $P(\text{STATE } 2 \mid A_{10}) = 0.63$), where values exceeding those expected by chance are indicated in red outline. A total of five out of six brain states were significantly associated with specific stimuli categories. Examination of *Figure 2A* can be used to provide a qualitative description of the estimated brain states. For example, STATE 2 was maximally observed when the circles were approaching and then collided, and thus reflects shock delivery (as well as peri-shock periods). As another example, STATE 5 was observed when the circles were in close proximity but did not touch. This state was particularly interesting given that the experiment was designed such that multiple 'near-miss' events would occur during the session, so as to enhance participants' state of apprehension when the circles approached each other.

We found that brain states were observed under specific stimulus conditions. But as the stimulus conditions change, do states transition in a systematic fashion? To evaluate this question, we determined the probability of a state transition given a specific stimulus category (*Figure 2B*, e.g., $P([\text{STATE } 2 \mapsto \text{STATE } 1] \mid A_{10}) = 0.88$; see Methods). *Figure 2B* shows state transitions that were significantly associated with a stimulus category. For example, STATE 2 (centered on the collision event) transitioned to STATE 1 ('post-shock') with very high probability for both stimulus categories $A_{10}$ and $R_{10}$ (i.e. when the circles collided). In *Figure 2B*, to facilitate understanding, we ordered the rows by starting with the STATE 1 to STATE 3 transition (from 'post-shock' to 'not close') and then chained the transitions in a manner that reflected the movement of the circles. Thus, the second row shows $\text{STATE } 3 \mapsto \text{STATE } 5$ (from 'not close' to 'near miss') and so on. For ease of reference, we summarize state descriptions in *Box 1*.

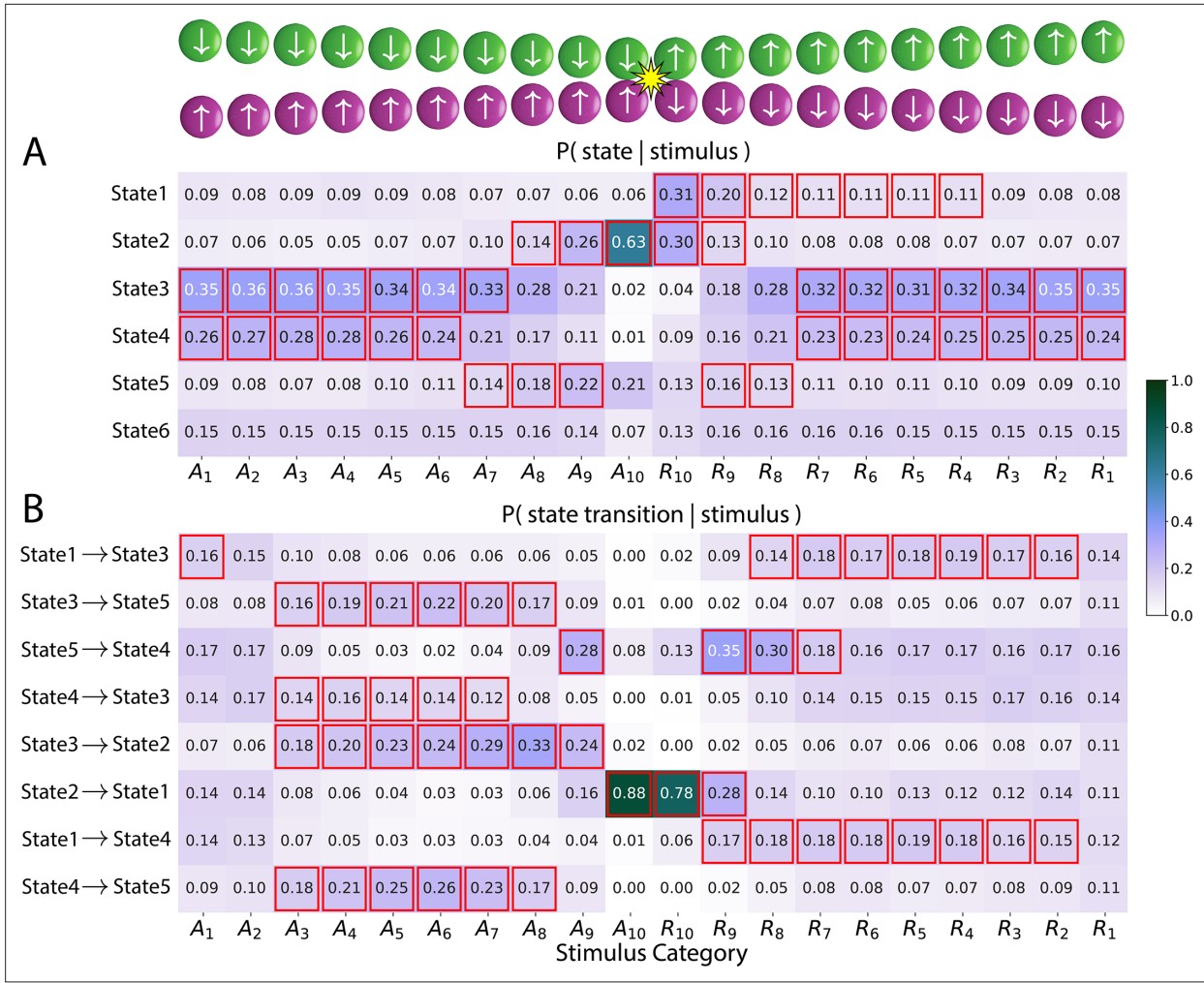

**Figure 2.** Brain states, state transitions, and input stimuli. Stimuli were categorized into 20 bins based on circle distance and movement direction: $A_1 - A_{10}$ for approach, $R_1 - R_{10}$ for retreat (1 indicates circles are farthest and 10 indicates circle collision). (**A**) Table entries indicate the probability of being in a state given the input stimulus category. (**B**) Table entries indicate the probability of a state transition given the input stimulus category. In both tables, cells highlighted in red indicate states/transitions significantly associated with the corresponding stimulus category ($p < 0.05$, corrected for multiple comparisons). The color scale indicates probability.

The online version of this article includes the following figure supplement(s) for figure 2:

**Figure supplement 1.** Brain states, state transitions, and input stimuli: version with 10 stimulus categories.

**Figure supplement 2.** Brain states, state transitions, and input stimuli: version with 30 stimulus categories.

Together, brain states and state transitions were systematically related to external stimuli, demonstrating that the model captured important properties of the experimental paradigm in an unsupervised manner.

## Model states capture threat processing

We reasoned that if states detected by the model reflect known mental states, they should be associated with brain activity with coherent spatial patterns consistent with those observed with standard fMRI designs (such as block and event-related designs). SLDS states and state transitions are defined in a latent space of dimensionality considerably lower than the original brain data space (here, 85 brain regions). Thus, to determine brain maps associated with specific states, we performed linear regression with a regressor that was on during the state in question and off otherwise (see Methods). *Figure 3* illustrates some of the states (see *Figure 3—figure supplement 1* for all states). For STATE 5, consistent with the 'near-miss' interpretation, multiple brain areas linked to increased anxious

## Box 1. SLDS state descriptions

- STATE 1: 'post-shock'. Observed right after the circles collided; only state that followed STATE 2 in *Figure 2B*.
- STATE 2: 'shock/peri-shock'. Observed during shock and peri-shock inputs.
- STATE 3: 'not near'. Observed when the circles were not near.
- STATE 4: 'not near'. Observed when the circles were not near and similar to STATE 3 in terms of inputs; STATE 4 followed STATE 5 when, after a near-miss collision, the circles continued to move apart from each other.
- STATE 5: 'near miss'. Observed when circles approached, came close to colliding, and then retreated.

apprehension were positively engaged, including the anterior insula and cingulate cortex (*Mobbs et al., 2009*; *Mobbs et al., 2010*; *Alvarez et al., 2011*; *Grupe and Nitschke, 2013*; *Somerville et al., 2013*; *Meyer et al., 2019*; *Hur et al., 2020*; *Murty et al., 2022*; *Murty et al., 2023*). STATE 4, which tended to follow STATE 5, exhibited a deactivation of these regions, consistent with the notion that participants experienced relative safety following a near collision (*Schiller et al., 2008*; *Grupe and Nitschke, 2013*; *Somerville et al., 2013*).

We further investigated the spatial properties of brain states by contrasting states. For example, the contrast of STATE 5 vs. STATE 4 revealed a pattern of activation very similar to that obtained in the literature for the contrast of higher vs. lower levels of threat. In particular, the contrast identified sectors of ventromedial PFC and posterior cingulate that have been linked to relative safety (*Schiller et al., 2008*; *Grupe and Nitschke, 2013*).

## Temporal evolution of activity

In the threat paradigm, as the circles moved continuously, we expected that brain signals would also vary temporally. If SLDS states captured meaningful information, state transitions should be associated with relevant activity change. To evaluate this possibility, we determined signal evolution during temporal windows centered around state transitions (see Methods). *Figure 4* shows signal evolution in the same order as the state transitions of *Figure 2B* in two key regions involved in threat-related processing, the dorsal anterior insula and the ventromedial PFC. For example, the transition from STATE 2 to STATE 1 reflected transitioning from a shock event to post-shock period. Accordingly, activity in the dorsal anterior insula, which is very sensitive to threat level (*Mobbs et al., 2009*; *Mobbs et al., 2010*; *Alvarez et al., 2011*; *Grupe and Nitschke, 2013*; *Somerville et al., 2013*; *Meyer et al., 2019*; *Hur et al., 2020*; *Murty et al., 2022*; *Murty et al., 2023*), underwent a drastic decrease. Likewise, when STATE 5 transitioned to STATE 4, activity in the dorsal anterior insula decreased,

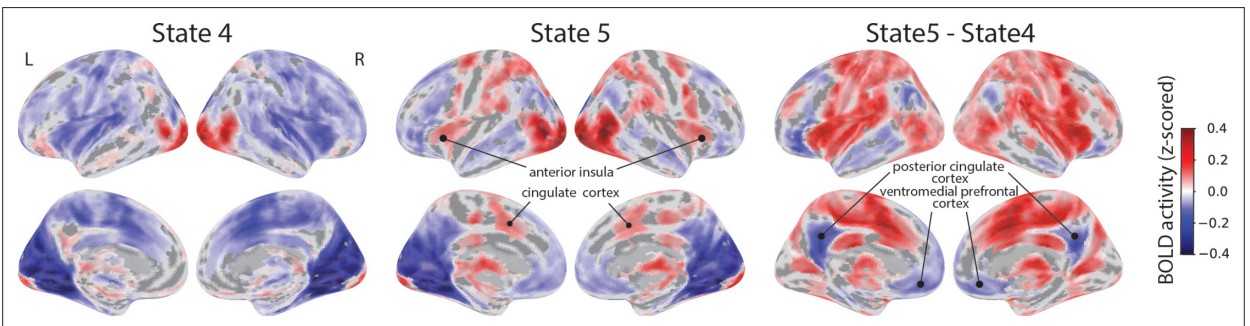

**Figure 3.** Voxelwise state activity maps and state contrast maps. State activity maps for STATE 4 and STATE 5 and the contrast of the two states. Maps were corrected for multiple comparisons by thresholding voxels at $p < 0.001$ and at the cluster level at $p < 0.05$.

The online version of this article includes the following figure supplement(s) for figure 3:

**Figure supplement 1.** Voxelwise state activity maps for all states.

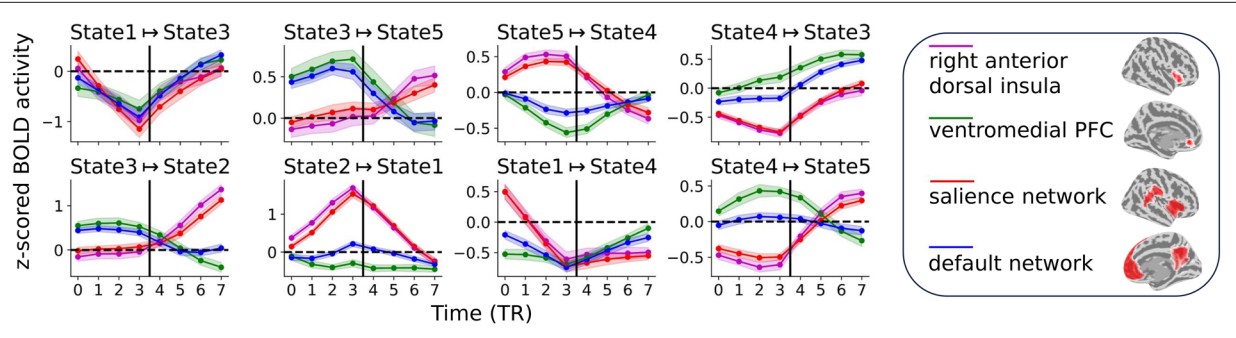

**Figure 4.** Region and network responses during state transitions. Vertical lines indicate the time of state transition. State transitions are arranged in a chained manner: STATE 1 ↦ STATE 3 ↦ STATE 5 ↦ STATE 4 ↦ STATE 3 ↦ STATE 2 ↦ STATE 1 ↦ STATE 4 ↦ STATE 5 to facilitate continuity in reading responses across state transitions. Error bars correspond to the 95% confidence interval based on the standard error of the mean across participants.

consistent with a transition from higher to lower threat and findings about responses in these two regions. In contrast, the transition from STATE 3 ('not close') to STATE 2 (peri-shock/shock) was associated with a sharp increase in activity strength in the dorsal anterior insula.

SLDS models states and state transitions in a latent space, which makes it possible to investigate activity dynamics in regions *not* used for estimating the models, because once a sequence of states is determined, one can examine activity in such regions at those times. Furthermore, activity can be determined at different spatial units, including ROI, voxel, or voxel cluster. In *Figure 4*, we probed the average signals from two resting-state networks engaged during threat-related processing, the salience network, which is particularly engaged during higher threat, and the default network, which is engaged during conditions of relative safety. As anticipated, signal evolution during the transition STATE 5 ↦ STATE 4 was similar to that observed for the dorsal anterior insula and the ventromedial PFC, although signal evolution for the default network deviated to some extent from that of the ventromedial PFC (note that the ROIs and networks discussed here spatially overlapped; they were not selected to be independent).

Overall, the model captured regularities of the experimental paradigm reflected in systematic changes in activity levels across the brain. For example, state transitions STATE 3 ↦ STATE 5, STATE 4 ↦ STATE 3, and STATE 4 ↦ STATE 5 were associated with approaching circles (increased threat) and increased responses in the dorsal anterior insula and salience network following the state transition. Similarly, the transition STATE 5 ↦ STATE 4 was associated with retreating circles (decreased threat) and increased responses in the ventromedial PFC and default network following the state transition. Furthermore, responses of the right anterior insula and salience network during STATE 2 ('shock/peri-shock') ramped up to the highest activity level with the STATE 3 ↦ STATE 2 transition from low to very high threat followed by a sharp decay with the STATE 2 ↦ STATE 1 transition that reflected the end of shock delivery.

## Threat dynamics

Each state is characterized by a state-specific trajectory in latent space. Such state dynamics are defined by an intrinsic component and an external-input component, which additively define the system's evolution (see *Figure 1* and Methods). In this section, we characterized the dynamics of threat processing based on state trajectories.

### Intrinsic state dynamics evolve toward fixed-point attractors

First, we examined *intrinsic* state dynamics, namely the component of the dynamics that is separate from external input contributions. Does intrinsic dynamics converge to an attractor, or does it diverge away from a fixed point? It is possible to test the stability of dynamics by evaluating properties of the dynamics matrix (matrix $A$; see Methods). We found that the intrinsic dynamics of all states studied were attractors (see *Figure 5—figure supplement 1*).

An attractor state corresponds to a point in latent space. Such a vector can be visualized as a map, which we call an 'attractor spatial map' by projecting it onto the original brain data space in terms of

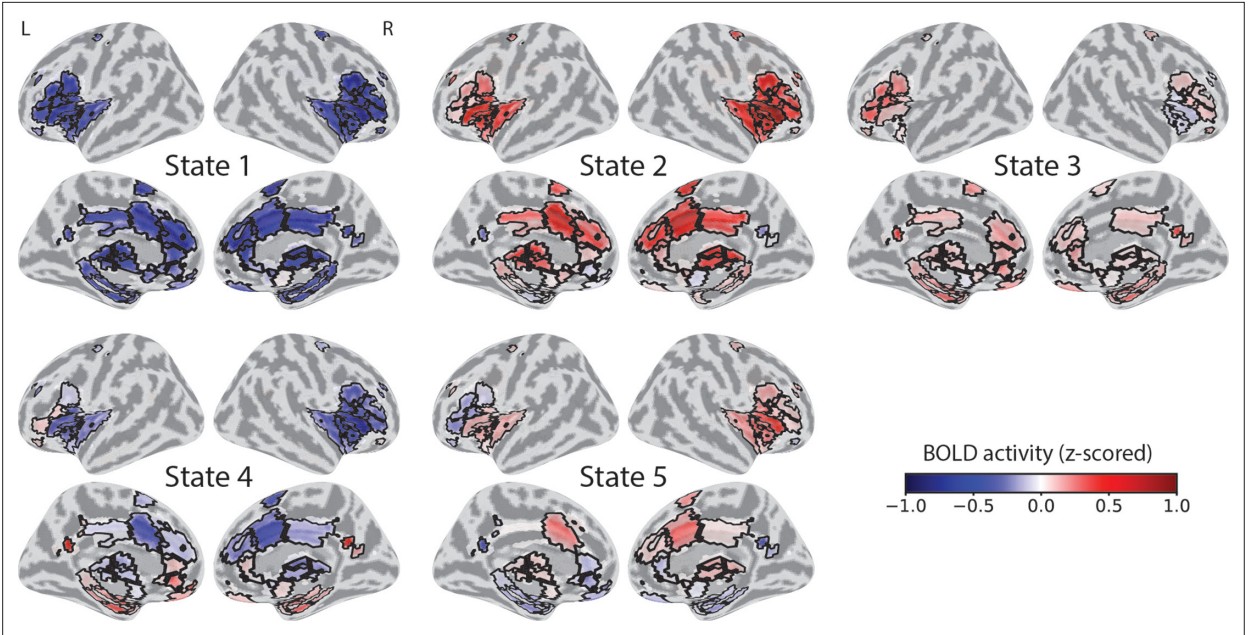

**Figure 5.** Attractor maps. State attractors projected onto the space of regions of interest (85 ROIs) visualized on a brain surface map. At each ROI, the color scale represents activity strength at the attractor's fixed point. ROI boundaries are marked in black.

The online version of this article includes the following figure supplement(s) for figure 5:

**Figure supplement 1.** Stability of states.

ROI activity on a brain surface (*Figure 5*; see Methods). For example, STATE 5 corresponded to 'near-miss' events and an attractor map with increased responses in the anterior insula/cingulate cortex and decreased responses in the ventromedial PFC and precuneus. The attractor for STATE 4 ('not near') exhibited an opposite organization consistent with the context of increased safety/lower threat.

Given that the SLDS model learns a generative model of the fMRI data during the experimental paradigm, state trajectories are based on both endogenous and exogenous contributions. How do these trajectories evolve relative to the state attractor? To investigate this question, we determined the vector field defined by the state's endogenous dynamics, where vectors represent the direction and magnitude of how a system evolves if started at specific points (*Figure 6*). Starting at a given point, trajectories evolved for multiple time steps in the overall direction of the attractor but after some time transitioned to other states (the green part of the trajectory indicates the mean lifetime of each state). The plots describe how, in a given state, the effective trajectory was determined by a combination of the endogenous flow and perturbations provided by the input. For example, in STATE

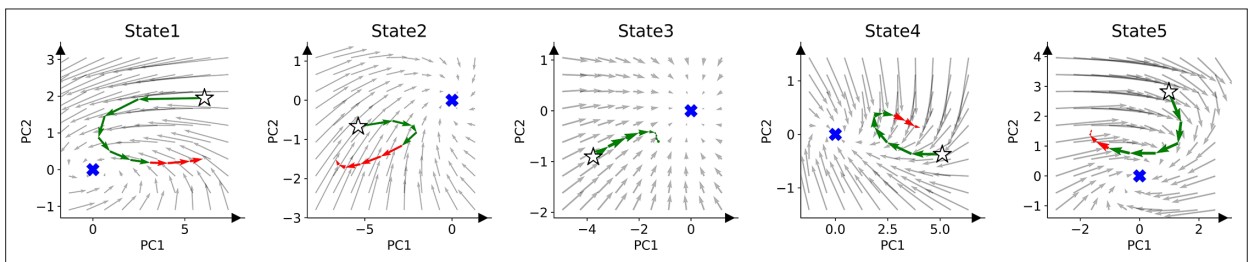

**Figure 6.** Evolution of state-specific trajectories. Trajectories were determined in the latent space and projected onto a two-dimensional vector field for illustration (coordinate axes are specific to each state). The average trajectory across participants, starting at the white star, is initially shown in green and switches to red when the majority of trajectories (across participants) switch to another state. Whereas the trajectory includes endogenous and exogenous contributions to temporal evolution, the vector field represents the endogenous contribution only. The gray arrows indicate the effect of the state's endogenous dynamics matrix, showing the direction and magnitude of evolution after a single time step. The blue cross indicates the state's fixed point attractor; the star indicates the start of the trajectory. PC: principal component.

2 (shock/peri-shock), the vector field exhibited less curvature than several other states, but trajectories were quickly steered away from the attractor because of inputs.

## Exogenous contributions to state trajectories

Following the qualitative description above (*Figure 6*), we next quantified the contributions of external inputs in steering system trajectories. For each state, we determined the state's centroid, namely, the centroid coordinate of all locations visited by the system during a specific state. Such geometrical location provided a reference position to investigate the input contributions. We considered the state centroid as a reference location because we sought to determine how the inputs steered trajectories relative to the 'most typical' position of a state.

We considered two scenarios. First, when the system was in state $i$, the input could drive the trajectory toward state $i$'s centroid or away from it (*Figure 7A*).

The 'effective contribution' was determined by considering both direction and magnitude components (see Methods). We investigated the stimulus categories defined previously ($A_1 - A_{10}$ and $R_{10} - R_1$). Consider STATE 1 (post-shock), for example (*Figure 7B*). When the stimulus was $R_{10}$ or $R_9$, those inputs tended to push the system in the direction of STATE 1's centroid; when the stimulus was $R_8$, the trajectory was pushed away from the centroid by the input but very weakly. In addition, for STATE 2 (shock/peri-shock), we found that stimuli $A_8$ to $R_{10}$ pushed the system toward the centroid, but stimulus $R_9$ moved the trajectory away from it. Overall, *Figure 7B* displays how inputs steered state trajectories in the direction of the centroid (green cells) or away from it (purple cells). Note that in *Figure 7B/C*, we evaluated exogenous contributions only for stimuli associated with each state/state-transition reported in *Figure 2A/B* (see also Methods).

Next, we investigated how inputs contributed to state transitions by steering trajectories in the direction of a new state's centroid (see Methods). Consider the STATE 2 $\mapsto$ STATE 1 transition (i.e. from shock to post-shock). Input perturbations $R_{10}$ and $R_9$ provided the strongest push of the trajectory in the direction of STATE 1. In addition, when the system was in STATE 3 ('not near'), multiple inputs tended to direct the system's evolution in the direction of STATE 2 ('shock/peri-shock'). Finally, when the system was in STATE 4 ('not close') and the circles were approaching, inputs pushed the system in the direction of STATE 5 ('near miss').

Taken together, the results show how external perturbations contributed to the evolution of state trajectories and state transitions.

## Region importance and steering of dynamics

Based on time series data and input information, the SLDS approach identifies a set of states and their dynamics. While these states are determined in the latent space, they can be readily mapped back to the brain, allowing for the characterization of spatiotemporal properties across the entire brain. Since not all regions contribute equally to state properties, we propose that a region's impact on state dynamics serves as a measure of its *importance*.

We illustrate the concept for STATE 5 ('near miss') in *Figure 8* (see *Figure 8—figure supplement 1* for all states). *Figure 8A* shows importance in the top row and activity below as a function of time from state entry. The dynamics of importance and activity can be further visualized (*Figure 8B*), where some regions of particularly high importance are illustrated together with the ventromedial PFC, a region that is typically not engaged during high-threat conditions. Notably, the importance of the dorsal anterior insula increased quickly in the first time points and later decreased. In contrast, the importance of the periaqueductal gray was relatively high from the beginning of the state and decreased moderately later.

*Figure 8C* depicts the correlation between these measures as a function of time. For all but STATE 1, the correlation increased over time. Interestingly, for STATES 4-5, the correlation was low at the first and second time points of the state (and for STATE 2 at the first time point), and for STATE 3 the measures were actually anticorrelated; both cases indicate a dissociation between activity and importance. In summary, our results illustrate that univariate region activity can differ from multivariate importance, providing a fruitful path to understand how individual brain regions contribute to collective dynamic properties.

## States explain threat processing in another threat-related experiment

If SLDS models capture spatiotemporal regularities associated with behavioral/brain states, determining states in a paradigm (such as the present moving circles paradigm) that shares important

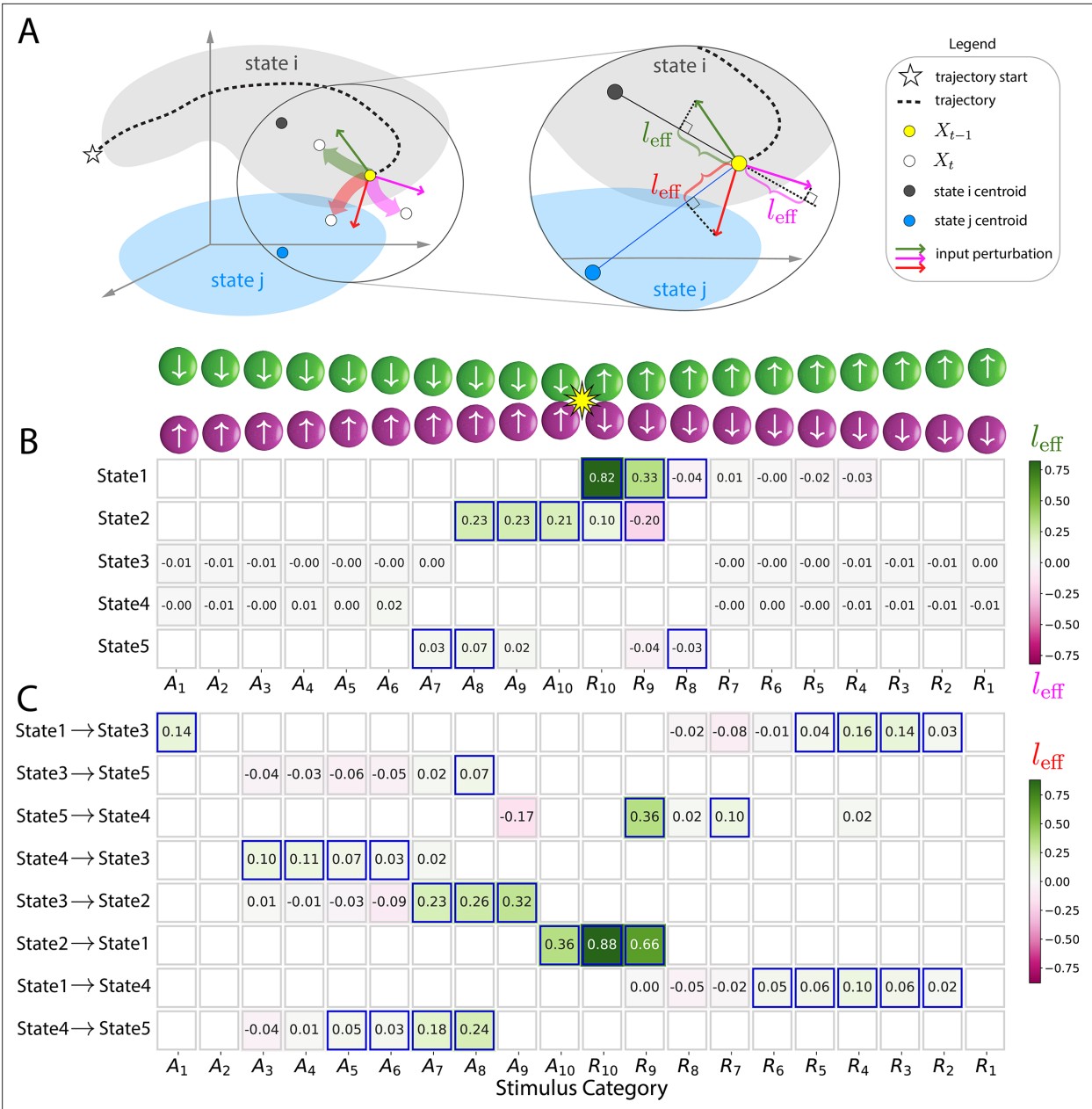

**Figure 7.** Effect of external inputs on steering state trajectories and driving state transitions. (**A**) Left: Trajectory at $t-1$ in state $i$. Inputs (colored arrows) could perturb the trajectory (translucent paths of the same color) by steering the evolution toward the state's centroid (green) or away from it (pink). When the input directed it away from the state's centroid, the input could push the system to switch into state $j$ (red). Right: The input effect was measured via the cosine of the angle $\phi$ between the input vector and the line joining $X_{t-1}$ and state centroids. (**B and C**) Input effects with rows and columns representing states/state-transitions and input categories, respectively. Only states with significant association with inputs and only significant state transitions are shown (see *Figure 2*). Cells with significant effects are highlighted in blue (p < 0.05, corrected for multiple comparisons).

The online version of this article includes the following figure supplement(s) for figure 7:

**Figure supplement 1.** Effect of external inputs on steering state trajectories and driving state transitions: version with 10 stimulus categories.

**Figure supplement 2.** Effect of external inputs on steering state trajectories and driving state transitions: version with 30 stimulus categories.

elements with a different experiment should be reflected when fitting the data of the latter. In other words, how well does a model estimated in one paradigm generalize to related experiments?

To test model generalizability, we applied the SLDS model estimated on the moving circles paradigm (call it MOV) to a separate experiment and dataset (call it ESC for 'escape') that investigated threat and safety processing. In the ESC experiment, participants manually controlled a rabbit icon

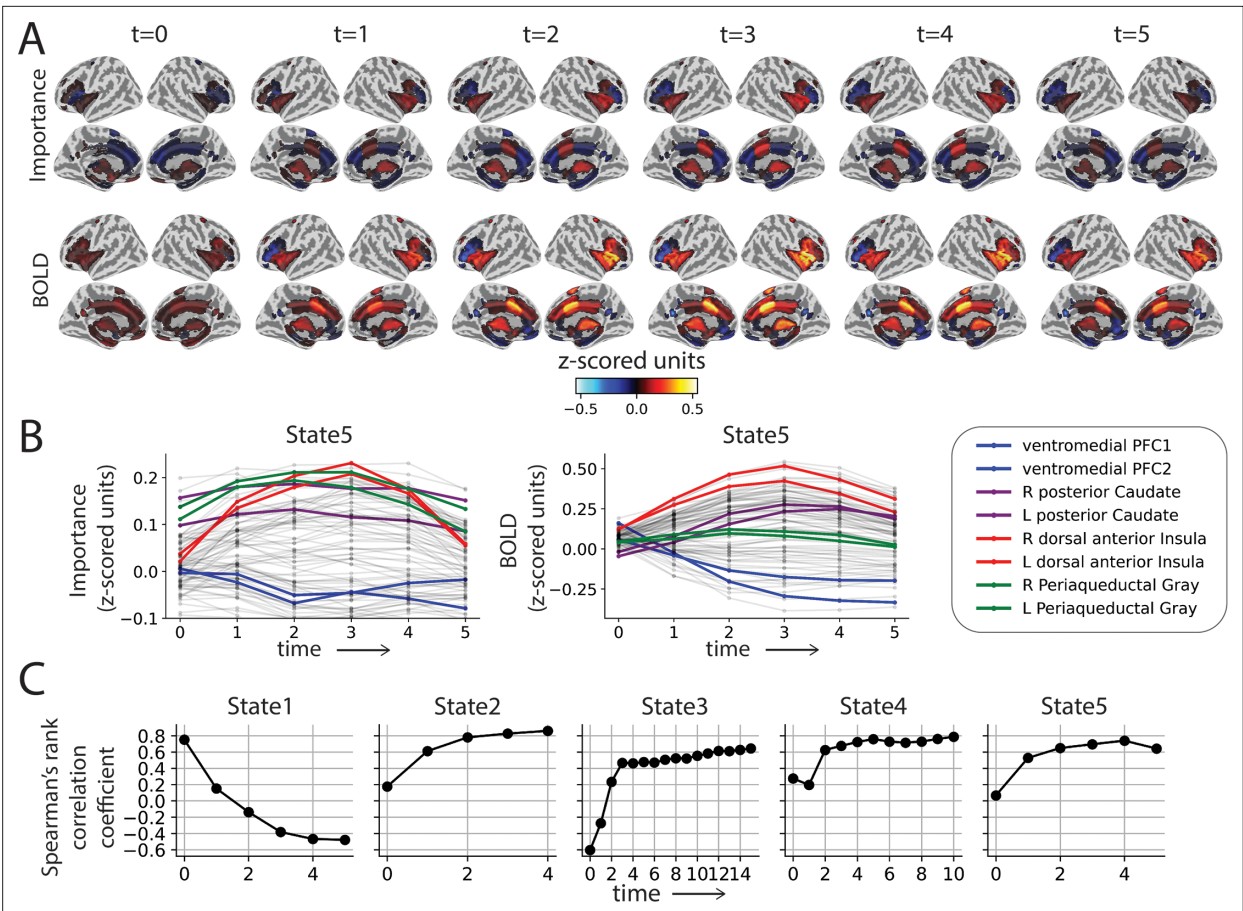

**Figure 8.** Brain region importance. (**A**) Importance measure and average BOLD signal for STATE 5 from state onset ($t = 0$) until the state's average lifetime. (**B**) Average importance measure time series (left) and average BOLD signal for all regions of interest (ROIs) during the average lifetime of STATE 5. (**C**) Spearman's rank correlation coefficient between average BOLD signal (across participants) and average importance measure (across participants) from state onset ($t = 0$) until the average lifetime of STATE 5. PFC: prefrontal cortex.

The online version of this article includes the following figure supplement(s) for figure 8:

**Figure supplement 1.** Importance measure of brain regions for all states.

using a joystick in a threat-avoidance task. The participant's goal was to reach a safety area on the opposite end of the screen while avoiding a pursuing devil icon moving at varying speeds. If caught, participants received a mild but unpleasant electrical stimulus. Once a gate was successfully crossed, the participant entered a safe area (no shock was possible) where they remained for a few seconds (see *Figure 9A*). Overall, the paradigm induced anxious apprehension during the chase phase and relief during safety.

Model parameters inferred from the MOV dataset were used to decode states in the ESC dataset. We evaluated the extent to which MOV states could explain fMRI time series data of the esc design. To do so, we focused on a 10-time-step-long temporal window anchored around the event of entering safety (from -6.25 s to 5.0 s, with $t = 0$, indicating the time of entering safety; see Methods). We determined the probability of being in a given MOV state given the time step (*Figure 9B*). The results indicated that STATE 5 ('near miss' in MOV) was expressed during the chase period; STATE 3 ('not near' in MOV) was detected from $t = 0$ s to the end of the temporal window. We also evaluated whether the system switched from STATE 5 to STATE 3 around $t = 0$ s by determining the probability of observing the transition STATE 5 ↦ STATE 3 as a function of time point (*Figure 9C*; see Methods).

In the MOV paradigm, STATE 5 was associated with a state of apprehension likely experienced by participants during near-miss events. This is consistent with the finding that STATE 5 was detected during the chase phase of the ESC paradigm. STATE 3 ('not near' in MOV) occurred during the safety

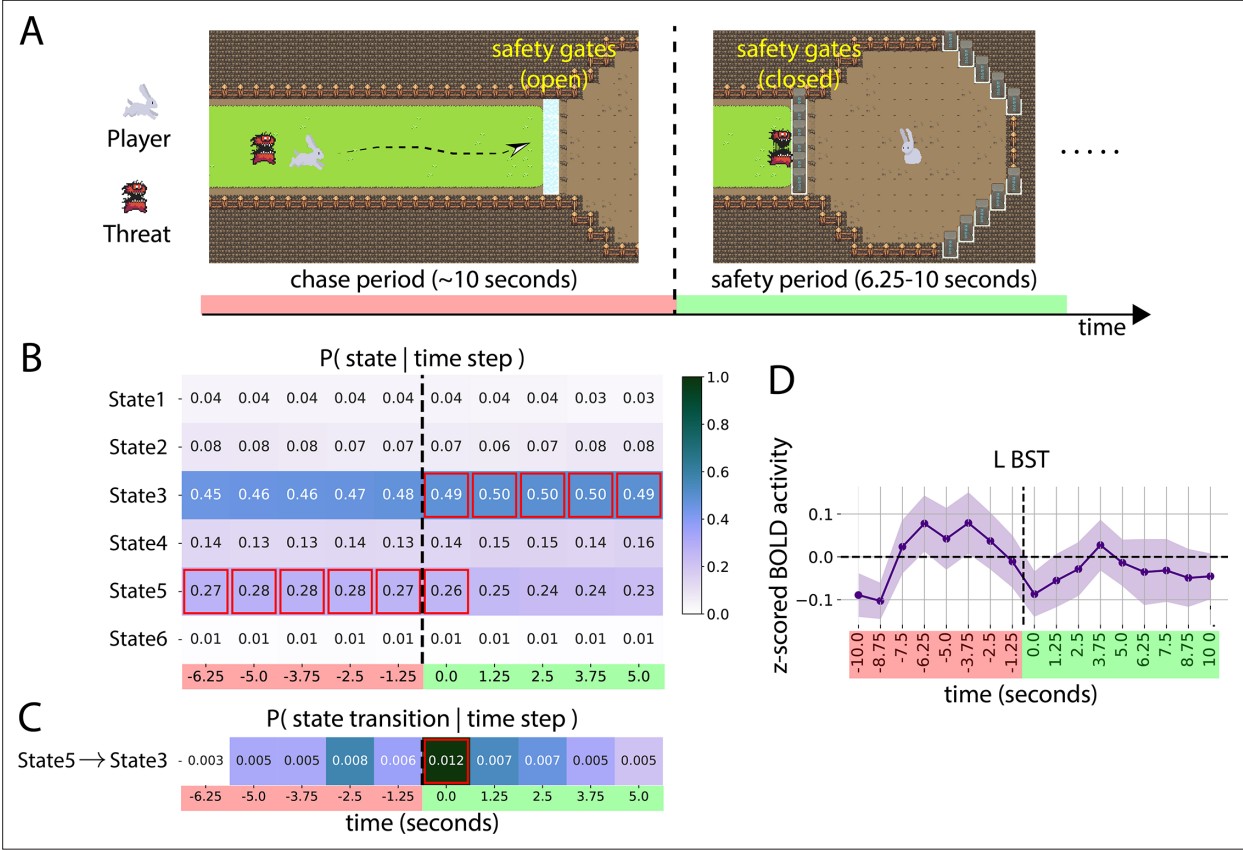

p < 0.05

**Figure 9.** Testing generalizability of the switching linear dynamical system (SLDS) model.

period (in ESC) and the transition from STATE 5 ↦ STATE 3 at $t = 0$. These findings indicate that the model trained on MOV was able to generalize in meaningful ways to the new paradigm.

To facilitate interpretation, no hemodynamic shifts were used in displaying the results shown in *Figure 9*. Our choice was motivated by the fact that the ESC paradigm contained a discrete event at $t = 0$, namely, the transition into safety. However, one concern is that STATE 3 was already detected at safety onset, whereas safety-related processing triggered at $t = 0$ should be expressed in fMRI responses with some lag given that any responses triggered at $t = 0$ should translate to fMRI responses with a lag of around 2 s and peak around 4–6 s post event (*Boynton et al., 1996*). Further analysis, however, helped explain the observed results, as outlined next. Average fMRI responses during the analyzed time window revealed a consistent response pattern across multiple brain regions sensitive to sustained threat in prior studies in the literature (*Mobbs et al., 2009*; *Davis et al., 2010*; *Murty et al., 2023*), as illustrated for the region of interest of the bed nucleus of the stria terminalis (BST; *Figure 9D*), a very important brain region involved in sustained threat processing (*Davis et al., 2010*). As shown, the BST response started decreasing before the end of the chase period, not at the end, suggesting that apprehension levels during the chase period decreased before the end of the chase period. To gain further insight concerning the response pattern of sustained-threat processing regions, we investigated the movement pattern of the chasing devil, which revealed a potential problem with the original experimental design. We programmed the chasing speed of the devil icon to be variable, but consistent across trials. In doing so, there was a consistent icon deceleration around the halfway point of the chase, which unintentionally made the final phase of the chase less apprehension-inducing. In all, we believe the expression of STATE 3 ('not near' in MOV) right at entry into the safety area, as well as the transition from STATE 5 ↦ STATE 3, is actually consistent with a mental state of relative safety, as also reflected in the activation pattern in the BST (and other brain regions).

## Discussion

In the present paper, we applied SLDSs to uncover the dynamics of threat processing during a continuous threat-of-shock paradigm. First, we demonstrated that the SLDS model learned the regularities of the experimental paradigm, such that states and state transitions estimated from fMRI time series data from 85 ROIs reflected both the proximity of the circles and their direction (approach vs. retreat). After establishing that the model captured key properties of threat-related processing, we characterized the dynamics of the states and their transitions. The results revealed that the understanding of threat processing is enhanced by viewing it in terms of dynamic multivariate patterns whose trajectories are a combination of intrinsic and extrinsic factors that jointly determine how the brain temporally evolves during dynamic threat.

Past work has employed state-space models such as HMMs to estimate states from brain signals in an unsupervised fashion. Previous fMRI studies detected states with blocked design (*Vidaurre et al., 2018*; *Yamashita et al., 2021*), where brain states are expected to align closely with the experimental structure. In continuous experimental paradigms, establishing brain states that meaningfully track experimental variables is considerably more challenging. In fact, in some studies employing HMMs with more unconstrained or naturalistic experiments, states were not explicitly mapped to stimulus events (*van de Meer et al., 2020*; *Song et al., 2021*; *Song et al., 2023*). A recent interesting study applied HMMs to cell recordings in the rat PFC and found states that correlated with hide-and-seek behaviors, as well as states corresponding to previously undetected behaviors (*Bagi et al., 2022*). In general, although HMM models applied to brain data have shown to be a promising strategy to uncover unlabeled states, further work is needed to assess their performance in continuous paradigms in a manner that rigorously establishes the correspondence between states and external stimuli and/ or behavior.

Critically, standard HMMs provide limited information about state dynamics, which is essential in understanding temporally unfolding brain processes supporting complex mental states and behaviors, such as the varying levels of threat and safety studied here. To address this gap, we applied the SLDS framework to our continuous threat-of-shock paradigm. There have been a few applications of SLDS models to brain data (*Petreska et al., 2011*; *Linderman et al., 2019*; *Glaser et al., 2020*), but as far as we know, only one fMRI study investigating a blocked working memory task (*Taghia et al., 2018*). As one of the challenges with unsupervised state-space methods is that there is no ground truth concerning the states, here, we first established that the model estimated states and transitions that captured the structure of the continuous paradigm, namely, circle proximity, shock events, and post-shock period. Of the six states detected by the model, five captured properties of the stimulus in a statistically significant manner. For example, the probability of being in STATE 2 given that the circles collided (input: $A_{10}$) was 0.63, consistent with a state indicative of a 'shock/peri-shock' period. State transitions were also associated with specific stimuli. More generally, the states captured the structure of the paradigm, which consisted of periods of increased and decreased threat given the proximity of the circles and the unpleasant shock delivered upon collision.

Further evidence that the states captured important features of threat-related processing was obtained by determining voxelwise brain maps for both individual states and state contrasts. For example, the contrast of STATE 5 ('near miss') vs. STATE 4 ('not close') revealed a pattern of activation very similar to that obtained in the literature for the contrast of higher vs. lower levels of threat (*Mobbs et al., 2009*; *Mobbs et al., 2010*; *Alvarez et al., 2011*; *Grupe and Nitschke, 2013*; *Somerville et al., 2013*; *Meyer et al., 2019*; *Hur et al., 2020*; *Murty et al., 2022*; *Murty et al., 2023*). In particular, the contrast identified sectors of ventromedial PFC and posterior cingulate that have been linked to relative safety (*Schiller et al., 2008*; *Somerville et al., 2013*; *Grupe and Nitschke, 2013*). In addition, state transitions reflected notable changes in activity levels, such as the transition of the lower-threat STATE 4 to the higher-threat STATE 5. Taken together, our analyses demonstrated that the model captured key properties of the continuous paradigm, building on previous studies using HMMs (*Vidaurre et al., 2018*; *van de Meer et al., 2020*; *Song et al., 2023*; *Baldassano et al., 2017*; *Williams et al., 2022*).

Our central goal was to discover and characterize the dynamics of continuous threat-related processing. From the perspective developed here, brain regions jointly participate in state-specific trajectories. An important property of the SLDS framework is that it estimates state dynamics based on separate and additive endogenous and exogenous contributions. Determining the intrinsic dynamics

is particularly informative because it uncovers how a brain state would evolve in the absence of additional inputs. In particular, does the state trajectory tend toward a fixed attractor, or does it evolve in some other more complex manner? Based on the properties of the endogenous state dynamics matrix, we determined that indeed all states behaved as attractors. But what do these attractors represent in terms of brain activity? To visualize the corresponding patterns, we determined 'attractor maps' by projecting the model attractor (determined in the latent space) onto the brain space (time series data from ROIs). The attractor maps captured essential properties of the states determined by the model. For example, STATE 1 and STATE 2 maps exhibited inverse activity patterns, consistent with the fact that they largely reflected the post-shock and the shock/peri-shock periods, respectively. In addition, comparison of STATE 5 ('near miss') and STATE 4 ('not near') shows how distributed patterns with shared dynamics are involved in mental states of higher and lower anxious apprehension.

The endogenous dynamics can be further understood as providing a flow field that channels the temporal evolution of the system in the absence of inputs. However, in our continuous threat-of-shock paradigm, state trajectories were also determined by the inputs acting at each time step. Thus, visualizing the trajectory in terms of vector fields uncovered both contributions to system behavior. In combination with the information obtained from attractor maps, these results provide novel ways to understand the roles of key brain regions involved in threat-related processing. In addition to the qualitative information provided by vector field trajectories, we quantified the input contributions to states and state transitions. We determined how inputs contributed to trajectories in a state's subspace and to transitions between states. For example, when the system was in STATE 2 (shock/peri-shock), the input perturbation $R_{10}$ steered the system in the direction of STATE 1 most vigorously, showing that soon after the circles collided, the brain tended to readily transition to a state of decreased activity in the insula and medial PFC. Overall, the analysis revealed how inputs steered a state's trajectory toward or away from the state's centroid, as well as how inputs directed a trajectory's evolution in the direction of a new state.

Work on dynamics of neural circuits in systems neuroscience at times assumes that the target circuit is driven only by endogenous processes (*Khona and Fiete, 2022*). In other words, the external inputs do not vary significantly over time and therefore do not perturb state trajectories. While such a scenario may be adequate in some experimental paradigms, the assumption is problematic in cases where inputs vary continuously. In the present study, we did not assume that the brain was isolated from the environment. Instead, by parsing the dynamics in terms of endogenous and exogenous contributions, we were able to not only study intrinsic state attractors but also to characterize external inputs as perturbations that drove trajectories along meaningful directions, such as pulling or pushing inputs toward or away from state centroids. Additionally, we also identified input stimuli that effectively steered evolution in the direction of other state subspaces. We propose that our approach can be profitably employed in studying neuronal circuit dynamics in systems neuroscience more broadly.

In the Introduction, we motivated our study in terms of determining multivariate and distributed patterns of activity with shared dynamics. At one end of the spectrum, it is possible to conceptualize the whole brain as dynamically evolving during a state; at the other end, we could focus on just a few 'key' regions or possibly a single one (at which point the description would be univariate). Here, we addressed this gap by studying the importance of regions to state dynamics: To what extent does a region steer the trajectory of the system? From a mathematical standpoint, our proposed measure is not merely a function of activity of a region but also of the coefficients of the dynamics matrix capturing its effect on across-region dynamics (*Eichler, 2005*; *Smith et al., 2010*).

How distributed should the dynamics of threat be considered? One answer to this question is to consider the distribution of importance values for all states. For STATE 1 ('post-shock'), a few regions displayed the highest importance values for a few time points. However, for the other states, the distribution of importance values tended to be more uniform at each time point. Thus, based on our proposed importance measure, we conclude that threat-related processing is profitably viewed as substantially distributed. Furthermore, we found that while activity and importance were relatively correlated, they could also diverge substantially. Together, we believe that the proposed importance measure provides a valuable tool for understanding the rich dynamics of threat processing. For example, we discovered that the dorsal anterior insula is important not only during high-anxiety states (such as STATE 5; 'near miss') but also, surprisingly, for a state that followed the aversive shock event (STATE 1; 'post-shock'). Additionally, we noted that posterior cingulate cortex, widely known to play

a central role in the default mode network, had the highest importance among all other regions in driving dynamics of low-anxiety states (such as STATE 3 and STATE 4; 'not near').

We believe the generalizability results presented in this study offer a valuable contribution to the ongoing discussion about the applicability of naturalistic paradigms in neuroscience. As highlighted in the Introduction, there is a growing interest in using more dynamic and ecologically valid experimental designs to advance our understanding of brain function. The successful application of the SLDS model, trained on the moving circles paradigm, to the separate escape experiment illustrates the potential of this approach to capture neural dynamics that generalize across related but distinct threat-processing tasks. This finding addresses a key challenge in naturalistic neuroscience: establishing that models derived from one paradigm can meaningfully explain brain activity in another. By showing that states identified in one threat-related task could be meaningfully mapped onto brain activity during a different threat-avoidance scenario, this study provides initial evidence that the SLDS framework can extract robust, task-general representations of threat processing, rather than idiosyncratic features of a particular experimental paradigm. Such cross-paradigm validity (see also *Song et al., 2023*) will be crucial for building a more comprehensive and ecologically relevant understanding of how the brain processes and responds to dynamic threats in real-world scenarios.

Here, we also discuss some challenges and potential shortcomings of our study. The application of state-space models, including SLDSs, to hemodynamic data is challenging because of the low-pass nature of the response. Specifically, the response to a brief event builds up to a peak around 4–6 s post onset and decays slowly over the next 4–8 s (*Boynton et al., 1996*). In the current study, we assumed a constant hemodynamic lag of approximately 5 s. Although our paradigm involved continuous movement of the circles, both shock events and transitions from approach to retreat (and vice versa) were discrete in nature. These events call for careful interpretation of the results given the potential for blending of hemodynamic responses. However, our experiment was designed such that most periods of circle approach and retreat lasted for 6–9 s so as to minimize consecutive approach/retreat transitions (*Limbachia et al., 2021*). Thus, the concern about the slow nature of the hemodynamic response was somewhat alleviated in the present context. In general, future studies are needed to develop SLDS approaches that can explicitly consider the properties of the hemodynamic response. Another limitation of the framework developed here is that we assumed, as in all state-model applications, that the underlying system is well approximated by a series of states. With SLDS models, this is less severe an assumption because one can consider the (possibly nonlinear) behavior of a system as modeled by a series of linear dynamical systems that approximate the overall behavior. Nevertheless, it will be valuable to investigate other approaches that address this concern, including considering that multiple states can coexist at a time (*Greene et al., 2023*). A further issue that we wish to discuss is related to the distinction between intrinsic and extrinsic dynamics, which is explicitly modeled in our SLDS approach (see Methods, *Equation 2*). We believe this is a powerful approach because in experimental designs with experimenter-manipulated inputs, one can profitably investigate both types of contribution to dynamics. However, complete separation between intrinsic and extrinsic dynamics is challenging to ascertain. More generally, one can gain confidence in their separation through controlled experiments where external inputs are manipulated, or by demonstrating time-scale separation of intrinsic and extrinsic dynamics.

In the present paper, we applied SLDSs to uncover the dynamics of threat processing during a continuous threat-of-shock paradigm. First, we demonstrated that the model learned the regularities of the experimental paradigm, such that states and state transitions estimated from fMRI time series data from 85 ROIs reflected both the proximity of the circles and their direction (approach vs. retreat). After establishing that the model captured key properties of threat-related processing, we characterized the dynamics of the states and their transitions. The results revealed that threat processing can profitably be viewed in terms of dynamic multivariate patterns whose trajectories are a combination of intrinsic and extrinsic factors that jointly determine how the brain temporally evolves during dynamic threat. We propose that viewing threat processing through the lens of dynamical systems offers important avenues to uncover properties of the dynamics of threat that are not uncovered with standard experimental designs and analyses.

## Methods

### Moving circles dataset

#### Participants

One hundred and twenty-six participants (63 females, ages 18–30 years; average: 20.87, STD: 2.56) with normal or corrected-to-normal vision and no reported neurological or psychiatric disease were recruited from the University of Maryland community. The project was approved by the University of Maryland College Park Institutional Review Board and all participants provided written informed consent before participation. Data from four participants were not used due to technical issues. For the original publication of these data, see *Limbachia et al., 2021*.

Out of the 122 participants, we used 30 to perform preliminary testing of the framework and defining model hyperparameters, including the number of states and the dimensionality of the latent space (see following subsections and Appendix 1). In order to avoid circularity in our analyses, data from this exploratory set were not used further. Thus, all results reported here are based on the held-out set of 92 participants.

#### Procedure and stimuli

The experiment was designed to study the role of proximity on threat-of-shock processing. Participants watched two circles of different colors on the screen, at times moving close to each other, at times moving apart in a smooth but unpredictable fashion. Upon circle collision, the participants received a mild but unpleasant electrical stimulation together with an aversive sound. The movement of the circles was defined such that there were segments of approach and retreat that lasted between 2 and 9 s, with most segments lasting more than 6 s. To enhance unpredictability and hence anxious apprehension by participants, the movement of the circles included multiple instances of 'near misses' in which the circles nearly touched (distance as close as 1.5 circle diameters between circle centers) before retreating away.

The experiment consisted of six 8 min runs, each having two 3 min blocks, during which the circles moved. The participants experienced, on average, 4 circle collisions and 7 near misses per run. All runs used different motion paths, but the visual stimulus was fixed across participants. Finally, visual stimuli were presented using PsychoPy (http://www.psychopy.org/) and viewed on a projection screen via a mirror mounted to the scanner's head coil. For further details about the experiment, the reader may refer to *Meyer et al., 2019*; *Limbachia et al., 2021*.

### Escape dataset

#### Participants

A total of 57 subjects (24 females, ages 18–54; average: 20.5, STD: 1.9) were recruited from the University of Maryland, College Park, community. All subjects had normal or corrected-to-normal vision and reported no neurological disease or current use of psychoactive drugs. They provided written informed consent before participating in the study and were paid immediately after the experiment. The study was approved by the University of Maryland, College Park, Institutional Review Board.

#### Procedure and stimuli

The experiment sought to study threat processing in the presence of 'imminent danger' followed by a period of safety. A total of eight 512-s-long runs were employed, each having 13.8±1.6 trials. Each trial began with a chase period that was followed by a safety period if the participant avoided being caught. Participants controlled a rabbit icon ('player'), which they could move on the screen at constant speed using a joystick. The player started at the left-hand side of the screen and their objective was to reach a safety area located at the right end of the screen while avoiding a devil icon that pursued the player at time-varying speeds. Participants received an unpleasant but not painful electrical stimulation if the devil caught the player before the player entered the safety region. If the player successfully entered the safety area, this period lasted 6–10 s.

### Regions of interest

Time series data used for both datasets consisted of fMRI time series from 85 cortical and subcortical ROIs (defined as the average across voxels for each ROI). The ROIs were defined in a prior study by

our group and intended to capture brain regions involved in threat-related processing (*Murty et al.,* *2022*).

## Switching linear dynamical systems

SLDSs are an example of a latent-variable framework that expresses the dynamics of observed signals using a set of unobserved variables assumed to govern system dynamics. The latent space is typically of considerably smaller dimension relative to the original data space (here 10 and 85, respectively). SLDSs consist of one linear dynamical system model per state, such that the trajectory of the system is governed by a single dynamical system at each time step. Switching between states is assumed to follow a Markov process. Overall, SLDS models provide a piecewise linear approximation of (potentially nonlinear) dynamics.

Consider activity $Y_t \in \mathbb{R}^N, t = 1, 2, \ldots, T$ ($T$ time steps is the duration of a typical run) from $N = 85$ brain ROIs during an experiment described using $M$ input variables $U_t \in \mathbb{R}^M, t = 1, 2, \ldots, T$. The model represents brain signals $Y_t$ in a low-dimensional subspace as a set of $D$ latent variables $X_t \in \mathbb{R}^D, t = 1, 2, \ldots, T$ ($D < N$). Observed BOLD activity is related to the latent variables using a linear transformation, such that

$$\left(Y_t | X_t\right) \sim \mathcal{N}\left(CX_t + d, \Sigma^v\right), \tag{1}$$

where $C \in \mathbb{R}^{N \times D}$ is the observation matrix, $d \in \mathbb{R}^N$ is a bias term, $\Sigma^v \in \mathbb{R}^{N \times N}$ is a noise covariance matrix, and $\mathcal{N}(\mu, \Sigma)$ is a multivariate Gaussian distribution with mean $\mu$ and covariance $\Sigma$. The expression above is called the observation step.

Latent trajectory dynamics is modeled using a set of $K$ linear dynamical systems that are indexed by a time-varying switching variable $Z_t \in \{1, 2, \ldots, K\}$, which we call the state. At each time step $t$, only the linear dynamical system corresponding to the state $Z_t$ drives the temporal evolution of system, such that

$$\left(X_t | X_{t-1}, Z_t, U_t\right) \sim \mathcal{N}\left(A_{Z_t} X_{t-1} + V_{Z_t} U_t + b_{Z_t}, \Sigma^h_{Z_t}\right), \tag{2}$$

where $A_{Z_t} \in \mathbb{R}^{D \times D}$ is the linear dynamics matrix, $V_{Z_t} \in \mathbb{R}^{D \times M}$ is the input matrix reflecting the experimental paradigm variables, $b_{Z_t} \in \mathbb{R}^D$ is a bias term, and $\Sigma^h_{z_t} \in \mathbb{R}^{D \times D}$ is a noise covariance matrix. The expression above is called the dynamics step.

State transition follows a modification of the Markov property, where the probability of switching to state $Z_t$ from the previous state $Z_{t-1}$ depends on the previous values of the state $Z_{t-1}$ and the latent variables at $t - 1$, $X_{t-1}$:

$$P\left(Z_t = j | Z_{t-1} = i, X_{t-1}\right) \propto \exp\left(\log(Q_{ij}) + r_j^\top X_{t-1}\right), \tag{3}$$

where $r_j \in \mathbb{R}^D$ and $Q \in R^{K \times K}$ is the Markovian transition matrix, whose $i$th row and $j$th column element indicates the transition probability from state $i$ to state $j$. This expression above is called the state switching step. $r_j$ is a weight vector (indexed by the state label, $j$) that specifies the additional dependency of $Z_t$ on $X_{t-1}$ (on top of the Markov dependence of $Z_{t-1}$ on $Z_t$), allowing state transitions based on the location of the trajectory in the latent space. SLDS with the above additional dependency is also known as recurrent SLDS (*Linderman et al., 2016*) and is shown to be more suitable to explain realistic time series than SLDSs without the recurrent dependencies (*Linderman et al., 2019*). Note that, on removing $r_j$ from the above equation, the state transition probability is the same as that of a Markov model.

A group-level SLDS model was estimated based on the data of 92 participants that participated in the moving circles experiment by using a variational Laplace Expectation-Maximization algorithm (*Linderman et al., 2016*). Specifically, all of the following were estimated: $C, d, \Sigma^v, \{A_k, V_k, b_k, \Sigma^h_k\}_{k=1}^K, Q, r$. In this manner, we obtained model variables $X_t$ and $Z_t$. We employed the SLDS implementation provided in the ssm Python library (https://github.com/lindermanlab/ssm, copy archived at *Linderman et al.,* *2025*).

## Modeling experimental stimuli as SLDS inputs

To encode the experimental paradigm, we categorized the stimuli based on the proximity and direction of motion of the moving circles. The instantaneous proximity was calculated as the normalized Euclidean distance between the circles, ranging from 0 (farthest) to 1 (touching). Direction was calculated based on the discrete difference of proximity: +1 (approach) or –1 (retreat). Proximity values were quantized into 10 equal-sized bins ($[0.0, 0.1), [0.1, 0.2), \cdots, [0.9, 1.0]$), one per binning per direction, resulting in 20 stimulus categories: $A_1, A_2, \cdots, A_{10}$ for approach; $R_1, R_2, \cdots, R_{10}$ for retreat. The stimulus conditions were thus encoded as a 20-dimensional binary vector $U_t$ at each time step $t$, with a 1 corresponding to the active category and 0 elsewhere. The number of stimulus categories was arbitrarily defined, but we verified that key results were robust to the number of input bins (see *Figure 2—figure supplement 1*; *Figure 2—figure supplement 2*; *Figure 7—figure supplement 1*, and *Figure 7—figure supplement 2*).

For a state $k$, the SLDS model represents the contribution of input on dynamics using the term $V_k U_t$ (*Equation 2*), where $U_t$ encodes the stimulus and $V_k$ maps the input into the latent space. Note $V_k$ is state-specific, allowing distinct representations of the same stimulus category per state.

## Assessing statistical significance using bootstrapping

Data from the held-out set of 92 participants of the moving circles dataset were used to report findings of the study and determine statistical significance, which was determined via bootstrapping (*Efron and Tibshirani, 1994*). A total of 500 bootstrap samples of the dataset were generated by sampling participants with replacement, and an SLDS model was fit on each bootstrap iteration.

Because state labels are not necessarily preserved across bootstrap samples, we realigned them using a combination of the Hungarian algorithm (*Kuhn, 1955*) and $k$-means clustering. To do so, for each bootstrap sample, state parameters $A_k, b_k, \Sigma_k^h$ were concatenated to form $K = 6$ 'large vectors' $\theta$, one for each state. The resulting vectors were then clustered into $K$ clusters. The $K$ cluster centroids provided representative vectors $\theta$ for each state. In this manner, aligning state labels across bootstrap samples amounted to assigning each bootstrap sample's $\theta$ vectors to the $K$ cluster centroids. The assignment problem was determined by the Hungarian assignment algorithm, such that the bootstrap sample's state $\theta$ vectors were maximally aligned to the $K$ cluster centroids.

To determine the number of bootstrap iterations, we generated samples in steps of 100 until the estimates of the above-mentioned $\theta$ centroids were stable. Specifically, the elements of each centroid did not change (within $10^{-5}$ precision) upon adding bootstrap samples. The procedure was stopped at 500 bootstrap samples.

## Calculating the probability of a state given a stimulus

To determine $P(\text{state} | \text{input})$, we defined a state by input table. Each entry corresponded to the proportion of time a state $Z_t$ assumed a specific value $z$ given that the input $U_t$ assumed a specific value $u$. This counting procedure can be briefly summarized in the (somewhat awkward) expression

$$P(Z_t = z | U_t = u) = \frac{\sum_{t=1}^{T} \mathbb{1}(Z_t = z)\mathbb{1}(U_t = u)}{\sum_{t=1}^{T} \mathbb{1}(U_t = u)},$$

where $\mathbb{1}(\cdot)$ represents an indicator function that takes value 1 when the operand is true and 0 otherwise. Given the hemodynamic delay (approximately 2 s to initiate a response and 4–6 s to peak; *Boynton et al., 1996*), we used a time shift of three time steps (3.75 s) in considering the relationship between SLDS state and stimulus category. That is, to facilitate interpretation, we assumed that the effect of circle proximity (as encoded in terms of the discrete stimulus categories) would be expressed in the fMRI time series 3 time steps later.

We tested the statistical significance of each entry in the state by input table as follows. To account for sampling variability of the sample we studied, we computed a table for each iteration of the bootstrap, allowing us to estimate the variability of our results across the population (group level). Similarly, we obtained surrogate tables, one for each bootstrap resample, by randomly shifting the state sequences by random offsets. In this manner, the relative order of the states was randomized with respect to order of stimuli but the other properties were maintained (transition structure in the sequences). By obtaining bootstrap and surrogate tables for each table cell corresponding to a given state and stimulus, we obtained a population-level distribution of table values and a null distribution.

Based on these results, each table entry mean was compared to the mean of the corresponding null distribution via a paired Wilcoxon signed-rank test at $p < 0.05$. To correct for multiple comparisons across the number of states and stimulus categories ($6 \times 20$), Bonferroni correction was applied.

## Calculating the probability of a state transition given a stimulus

State transitions were linked to the stimuli using an analogous procedure as that to map states to stimuli. We created a table with 8 state transitions of interest (see Appendix 1) by 20 stimulus categories. Each table entry was calculated by counting the number of times a given state transition was observed given a stimulus category. Formally, this counting procedure can be succinctly described as follows: for a row $r$ representing transition $\text{state } i \mapsto \text{state } j$ and column $u$, the corresponding table entry measures the probability

$$P(Z_t = i, Z_{t+1} = j | U_t = u) = \frac{\sum_{t=1}^{T} \mathbb{1}(Z_t = i)\mathbb{1}(Z_{t+1} = j)\mathbb{1}(U_t = u)}{\sum_{t=1}^{T} \mathbb{1}(U_t = u)},$$

where $\mathbb{1}(\cdot)$ represents an indicator function. Once again, this is calculated by considering the hemodynamic delay. Finally, statistical inference was performed using a similar approach as described previously for calculating probabilities of states given stimuli.

## State spatial maps

We sought to characterize the spatial organization of brain states: for a given state $k$, what is the typical pattern of activity throughout the brain. This could be accomplished by using multiple regression with variables for each state: 1 when the state is on, 0 otherwise. We performed multiple regression in this manner in a voxel-wise manner. Note that this was possible because once the states are estimated (based on 85 ROIs), the regression can be applied at any spatial level desired (ROI or voxel). The regression model can be expressed mathematically as

$$\text{BOLD}_t = \sum_{i=k}^{K} \beta_i \times \mathbb{1}(Z_t = i) + \varepsilon_t,$$

where $\mathbb{1}(Z_t = k)$ is an indicator function denoting whether state $k$ is active at time $t$, $\beta_k$ is the corresponding regression coefficient, and $\varepsilon_t$ is an error term. For notational simplicity, terms accounting for confounds like scanner drift and head motion have been omitted from the above expression but were included in the regression model.

State maps were evaluated statistically based on the bootstrap approach discussed previously. First, state activity maps were determined for every participant, separately. A group-level map was then obtained by averaging across participants. This procedure was performed for every bootstrap sample, allowing us to estimate sampling variability. Similarly, a set of null state maps was generated by repeating the same procedure but with randomly permuted state sequences. To identify statistically significant voxels, we performed a paired t-test between the participant-based bootstrapped maps and the null maps. To account for multiple comparisons, we employed a cluster-based approach (*Friston et al., 1994*). Voxels were first thresholded at $p < 0.001$, and clusters with at least $c = 18$ voxels were deemed statistically significant at the cluster level. To determine the minimum cluster size $c$, we employed null state maps. The cluster size $c$ was such that only 5% of the clusters in the null state maps would be detected, corresponding to a 5% false-positive rate.

For contrast maps, an analogous approach was employed, but at the individual level, contrast maps were determined by computing $\beta_i - \beta_j$, where $i$ and $j$ are two states in question.

## Responses during state transitions

Responses at a given ROI were evaluated during temporal windows centered around state transitions. For transitions from state $i$ to state $j$, we chose windows where the first half of the time steps were labeled as state $i$ and the final time steps as state $j$. We chose windows of length 8 time steps such that they were long enough to reveal activity changes but also observed frequently enough to generate a reliable estimate. For a given state transition, participant-level responses were determined by averaging responses across all qualifying time windows. To estimate variability, group-level responses were determined by averaging participant-level responses for each bootstrap sample.

We also determined responses at the 'network' level, which were obtained by averaging ROI responses across all ROIs inside the network. We considered the salience (ventral attention) and default networks, as defined in the Schaefer 100-region 7-network parcellation (*Schaefer et al., 2018*).

## State-specific intrinsic dynamics and fixed points

Each state's dynamics in the latent space was modeled using a linear dynamical system, which evolves as follows:

$$X_t = A_k X_{t-1} + b_k + V_k U_t. \tag{4}$$

To characterize the intrinsic dynamics exclusively, assume that the input to the system is zero across time:

$$X_t = A_k X_{t-1} + b_k. \tag{5}$$

Assuming that the dynamics are stable and approach an equilibrium point $\overline{X}_k$ as a function of time (also known as a 'fixed point' or attractor), we obtain

$$\overline{X}_k = A_k \overline{X}_k + b_k$$
$$\overline{X}_k = [I - A_k]^{-1} b_k,$$

where $I$ is a $D \times D$ identity matrix. All trajectories in the neighborhood of the fixed point (also called the basin of attraction) will eventually converge toward the fixed point.

## Testing for stable intrinsic dynamics

The dynamics of a linear dynamical system is determined by the eigenvalues of the dynamics matrix $A_k$. In particular, the dynamics is stable if all eigenvalues lie inside a unit circle on the complex plane (refer to *Luenberger, 1979*; *Robinson, 2012*, for detailed accounts on stability of linear dynamical systems). Therefore, it suffices to test whether the norm of the largest eigenvalue is less than unity. Bootstrapping was again used to evaluate the statistical significance of the results. We determined the largest eigenvalues for all bootstrap samples and calculated the p-value as the fraction of the eigenvalues with absolute value greater than 1. States with p-value<0.05 were considered to follow stable dynamics. Bonferroni correction was applied to correct for multiple comparisons across $K$ states.

## Visualizing state attractors on the brain

Given an attractor state $i$, its fixed point $\overline{X}_i$ is a vector in the $D$-dimensional latent space. For the purpose of visualizing state attractors on the brain, we projected them onto the original data space (the time series space of 85 ROIs) using the observation step of the SLDS model (*Equation 1*).

$$\overline{X}_i^{\mathrm{ROI}} = C\overline{X}_i + d \qquad i = 1, 2, \cdots, K,$$

where $\overline{X}_i^{\mathrm{ROI}}$ is state $i$'s attractor in ROI space. State attractors in ROI space were estimated across bootstrap samples, and the average pattern was visualized. The average attractor pattern was thresholded at each ROI ($p < 0.05$) based on a Wilcoxon signed-rank test ($p < 0.05$), i.e., by comparing the activity levels to zero. Bonferroni correction was applied to correct for multiple comparisons across $K$ states and 85 ROIs.

## State trajectories and vector fields

For each state, trajectories were determined first in the latent space ($D = 10$). We averaged (across participants) all windows with 10 time steps starting at a specific state $k$. Note that trajectories correspond to *Equation 4* and consider both endogenous and exogenous terms. At each time step of the average trajectory, a trajectory could continue along a state or switch to a state $j \neq k$. The state label for a time step along the trajectory was assigned to the most typical state at that time step, i.e., the modal state. We colored the trajectory green when the state label was $k$ and red otherwise (state $j \neq k$). To illustrate the trajectory, we plotted it in 2D. To do so, we fit principal component analysis on the 10 points constituting the average trajectory and projected it onto the plane spanned by the top two principal components.

Trajectories were plotted on vector fields that represent how a particle would flow under the endogenous dynamics (i.e. based on $A_k$). This allowed us to visualize how external inputs contributed to the evolution of the system (note that, by definition, without inputs trajectories would converge to the state's attractor). At each point on the 2D plane, the vectors represent the trajectory evolution for a single time step. Vector field points were sampled uniformly on the 2D plane. The points were projected to the latent space using the top two principal components above, evolved according to endogenous dynamics rule (*Equation 5*), and projected back onto the 2D plane again using the top two principal components. Finally, the state's fixed point attractor (a point in the latent space) was also projected onto the 2D plane in the same manner.

## Quantifying contribution of inputs in state dynamics

As described above, state $k$'s trajectory follows the following dynamical rule given in *Equation 4*, with the term $V_k U_t$ representing contribution of external inputs as a vector in latent space. To investigate the contribution of external inputs to the latent trajectory, we determined how inputs steered the system's trajectory. For each state, we determined the state's centroid, namely, the vector $X^{c,k}$ that represented the centroid coordinate of all locations visited by the system during state $k$. Such geometrical location provided a reference position to investigate the input contributions.

Given a state, the external input at a given time can either drive latent trajectories toward the state centroid or away from it. To quantify input effects, we measured the length of the projection of the input vector along the direction connecting the position of the state trajectory and the state centroid:

$$l_{\text{eff}} = \frac{\langle X^{c,k} - X_t, V_k U_t \rangle}{\| X^{c,k} - X_t \|},$$

where $\langle \cdot, \cdot \rangle$ indicates the inner product between two vectors and $\| \cdot \|$ indicates the Euclidean norm. Thus, positive (negative) $l_{\text{eff}}$ values indicate that the input pushes state trajectories toward (away from) the state centroid. We focused on inputs associated with the given state as defined throughout the paper (methods for mapping states to stimuli, see Methods). Accordingly, the input effects are summarized in a table where the $(i,j)$th element represents the effect of input $j$ on trajectories in state $i$.

Tables were constructed as previously such that each bootstrap iteration generated one instance of the table. The results shown correspond to the average value across bootstrap samples. Statistically, we tested whether the average input effects were significantly different from zero by conducting a Wilcoxon signed-rank test at each table entry ($p < 0.05$). Bonferroni correction was applied to correct for multiple comparisons across $K = 6$ states and 20 input categories.

## Quantifying the contribution of inputs in state transitions

Next, we quantified the contributions of input stimuli to state transitions. Specifically, for particular state transitions $i \mapsto j$, we determined the extent to which input stimuli contributed to such transitions. As above, $l_{\text{eff}}$ was used to measure the effect of the external input vector along the direction joining the position of the state trajectory and state $j$'s centroid (in place of state $i$'s centroid). Positive values indicate that the input pushed the latent vector from state $i$ toward state $j$.

Tables were constructed as previously, such that each bootstrap iteration generated one instance of the table. The results shown correspond to the average value across bootstrap samples. Statistically, we tested whether average input effects were significantly different from zero by conducting a Wilcoxon signed-rank test at each table entry ($p < 0.05$). Bonferroni correction was applied to correct for multiple comparisons across 8 state transitions of interest and 20 input categories.

## Testing SLDS model generalizability

After fitting the SDLS model based on the data of the moving circles (MOV) paradigm, we investigated if the model states generalized to another threat-related paradigm, namely, the escape (ESC) experiment described above (see *Figure 9A*). Specifically, after fitting the data to the first dataset, we determined the variables $X_t$ and $Z_t$ for the ESC dataset. ESC time series data was analyzed for temporal windows centered around the time step when the player entered safety (rounded to the nearest data sample). Windows straddled between chase and safety periods, from $t = -6.25$ to $t = 5.0$ with $t = 0$ being the cross-into-safety point.

To determine the best-matching MOV states for the ESC time series data, we quantified $P(\text{state}|\text{time step})$ for all combinations of state by time steps (see **Figure 9**). Each value was computed as the normalized frequency of observing a state $Z_t$ at a given time step $t_{\text{win}}$ of the window. Formally:

$$P(Z_t = z|t_{\text{win}} = i) = \frac{\sum_{t=1}^{T} \mathbb{1}(Z_t = z)\mathbb{1}(t_{\text{win}} = i)}{\sum_{t=1}^{T} \mathbb{1}(t_{\text{win}} = i)},$$

where $\mathbb{1}(\cdot)$ represents the indicator function.

We determined the statistical significance of the probability values as follows. To determine the probabilities that would be observed by chance, we determined surrogate tables (states by time step) by shifting the MOV state sequences by random offsets (analogous to the approach used in mapping states to stimuli in MOV, see Methods), which provided a null distribution of probability values. Each probability value was then compared to the mean of the corresponding null distribution via a paired Wilcoxon signed-rank test at p < 0.05. To correct for multiple comparisons across the number of states and time steps (6×10), Bonferroni correction was applied. In a similar fashion, the probability of a state transition from $\text{STATE}\,5 \mapsto \text{STATE}\,3$ was tested for statistical significance (analogous to mapping state transitions to stimuli in MOV, see Methods).

## Region importance

We performed a 'lesion study', where we quantified how brain regions contribute to state dynamics by eliminating (zeroing) model parameters corresponding to a given region and observing the resulting changes in system dynamics. According to our approach, the most important regions are those that cause the greatest change in system dynamics when eliminated.

The SLDS model represents dynamics in a low-dimensional latent space, and model parameters are not readily available at the level of individual regions. Thus, the first step was to project the dynamics equation onto the brain data prior to computing importance values. Thus, the linear dynamics equation in the latent space (**Equation 2**) was mapped to the original data space of $N = 85$ ROIs using the emissions model (**Equation 1**):

$$Y_t = CX_t + d$$

$$= C\left(A_{Z_t}X_{t-1} + V_{Z_t}U_t + b_{Z_t}\right) + d$$

$$= CA_{Z_t}C^\dagger CX_{t-1} + CV_{Z_t}U_t + Cb_{Z_t} + d$$

$$= CA_{Z_t}C^\dagger\left(CX_{t-1} + d - d\right) + CV_{Z_t}U_t + Cb_{Z_t} + d$$

$$= \underbrace{\left(CA_{Z_t}C^\dagger\right)}_{\equiv \widehat{A_{Z_t}}}Y_{t-1} + \underbrace{\left(CV_{Z_t}\right)}_{\equiv \widehat{V_{Z_t}}}U_t + \underbrace{\left(Cb_{Z_t} + \left(I - CA_{z_t}C^\dagger\right)d\right)}_{\equiv \widehat{b_{Z_t}}}$$

$$= \widehat{A_{Z_t}}Y_{t-1} + \widehat{V_{Z_t}}U_t + \widehat{b_{Z_t}},$$

where $C^\dagger$ represents the Moore-Penrose pseudoinverse of $C$, and $\widehat{A_{Z_t}} \in \mathbb{R}^{N \times N}$, $\widehat{V_{Z_t}} \in \mathbb{R}^{N \times M}$, and $\widehat{b_{Z_t}} \in \mathbb{R}^{N}$ denote the corresponding dynamics matrix, input matrix, and bias terms in the original data space.

Based on the above, we defined the importance of the $i$th ROI at time $t$ based on quantifying the impact of 'lesioning' the $i$th ROI, i.e., by setting the $i$th column of $\widehat{A_{Z_t}}$, the $i$th row of $\widehat{V_{Z_t}}$, and the $i$th element of $\widehat{b_{Z_t}}$ to 0, denoted $\widehat{A_{Z_t}}^{(-i)}$, $\widehat{V_{Z_t}}^{(-i)}$, and $\widehat{b_{Z_t}}^{(-i)}$, respectively. Formally, the importance of the $i$th ROI $q_t^i$ was defined as:

$$q_t^i = \left\|\left(\widehat{A_{Z_t}}Y_{t-1} + \widehat{V_{Z_t}}U_t + \widehat{b_{Z_t}}\right) - \left(\widehat{A_{Z_t}}^{(-i)}Y_{t-1} + \widehat{V_{Z_t}}^{(-i)}U_t + \widehat{b_{Z_t}}^{(-i)}\right)\right\|_2$$

$$= \left\|\left(\widehat{A_{Z_t}}Y_{t-1} - \widehat{A_{Z_t}}^{(-i)}Y_{t-1}\right) + \left(\widehat{V_{Z_t}}U_t - \widehat{V_{Z_t}}^{(-i)}U_t\right) + \left(\widehat{b_{Z_t}} - \widehat{b_{Z_t}}^{(-i)}\right)\right\|_2$$

$$= \left\|Y_{t-1}^i * \widehat{A_{Z_t}}^{(i)} + \left(\widehat{V_{Z_t}}^{(i)}U_t\right) * \mathbb{1}_i + \widehat{b_{Z_t}}^{(i)} * \mathbb{1}_i\right\|_2,$$

where '$*$' indicates element-wise multiplication of a scalar with a vector, $Y_{t-1}^i$ is the activity of $i$th ROI at time $t - 1$, $\widehat{A_{Z_t}}^{(i)}$ corresponds to the $i$th column of $\widehat{A_{Z_t}}$, $\left(\widehat{V_{Z_t}}^{(i)\top}U_t\right)$ is the inner product between $i$th

row of $\widehat{V_{Z_t}}$ and input $U_t$, $\widehat{b_{Z_t}}^{(i)}$ corresponds to the $i$th element of $\widehat{b_{Z_t}}$, and $\mathbb{1}_i \in \mathbb{R}^N$ represents an indicator vector corresponding to the $i$th ROI. Note that the term $Y_{t-1}^i * \widehat{A_{Z_t}}^{(i)}$ is a function of both the $i$th ROI's activity and the coefficients of the dynamics matrix capturing the effect of region $i$ on the one-step dynamics of the entire system (*Eichler, 2005*; *Smith et al., 2010*); the remaining terms capture the effect of the external inputs and the bias term on the one-step dynamics of the $i$th ROI.

After computing $q_t^i$ for a given run, the resultant importance time series was normalized to zero mean and unit variance.

## Acknowledgements

This research was supported by the National Institute of Mental Health (R01-MH-071589). We thank Xiaoyu Zhou for discussions about the statistical approach developed in the paper.

## Additional information

### Funding

| Funder | Grant reference number | Author |
| --- | --- | --- |
| National Institute of Mental Health | R01-MH-071589 | Luiz Pessoa |

The funders had no role in study design, data collection and interpretation, or the decision to submit the work for publication.

### Author contributions

Joyneel Misra, Software, Formal analysis, Validation, Investigation, Visualization, Methodology, Writing – original draft; Luiz Pessoa, Conceptualization, Resources, Data curation, Supervision, Funding acquisition, Investigation, Writing – original draft, Project administration, Writing – review and editing

### Author ORCIDs

Joyneel Misra (ID) https://orcid.org/0000-0001-8306-4832

### Ethics

Escape Dataset: A total of 57 subjects (24 females, ages 18-54; average: 20.5, STD: 1.9) were recruited from the University of Maryland, College Park, community. All subjects had normal or corrected-to-normal vision and reported no neurological disease or current use of psychoactive drugs. They provided written informed consent before participating in the study and were paid immediately after the experiment. The study was approved by the University of Maryland, College Park, Institutional Review Board. Moving Circles Dataset: One hundred and twenty-six participants (63 females, ages 18-30 years; average: 20.87, STD: 2.56) with normal or corrected-to-normal vision and no reported neurological or psychiatric disease were recruited from the University of Maryland community. The project was approved by the University of Maryland College Park Institutional Review Board and all participants provided written informed consent before participation.

Reviewer #1 (Public review): https://doi.org/10.7554/eLife.102539.3.sa1
Reviewer #2 (Public review): https://doi.org/10.7554/eLife.102539.3.sa2
Author response https://doi.org/10.7554/eLife.102539.3.sa3

## Additional files

### Supplementary files

MDAR checklist

## Data availability

All data and code for the analyses presented in this paper are openly accessible at https://github.com/codejoydo/moving_circles_SLDS, (copy archived at *Misra, 2025*).

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

## Appendix 1

### Determining the number of latent dimensions

Data from an exploratory set of participants (*N*=30) was used to choose the dimensionality of the latent space. To do so, we computed the evidence lower bound (ELBO) as the optimization criterion of the SLDS fits to the data (*Kingma and Welling, 2022*; *Linderman et al., 2019*). Intuitively, the `ELBO` is a lower bound on the log-likelihood of the observed data, therefore capturing the goodness of the model fit (i.e. the ability to explain the observed data). For each value, an SLDS model was fit on 30 leave-one-out participant samples, and the average `ELBO` value across leave-one-out samples was determined. `ELBO` values in the range were very similar according to the figure below, so we chose *D*=10.

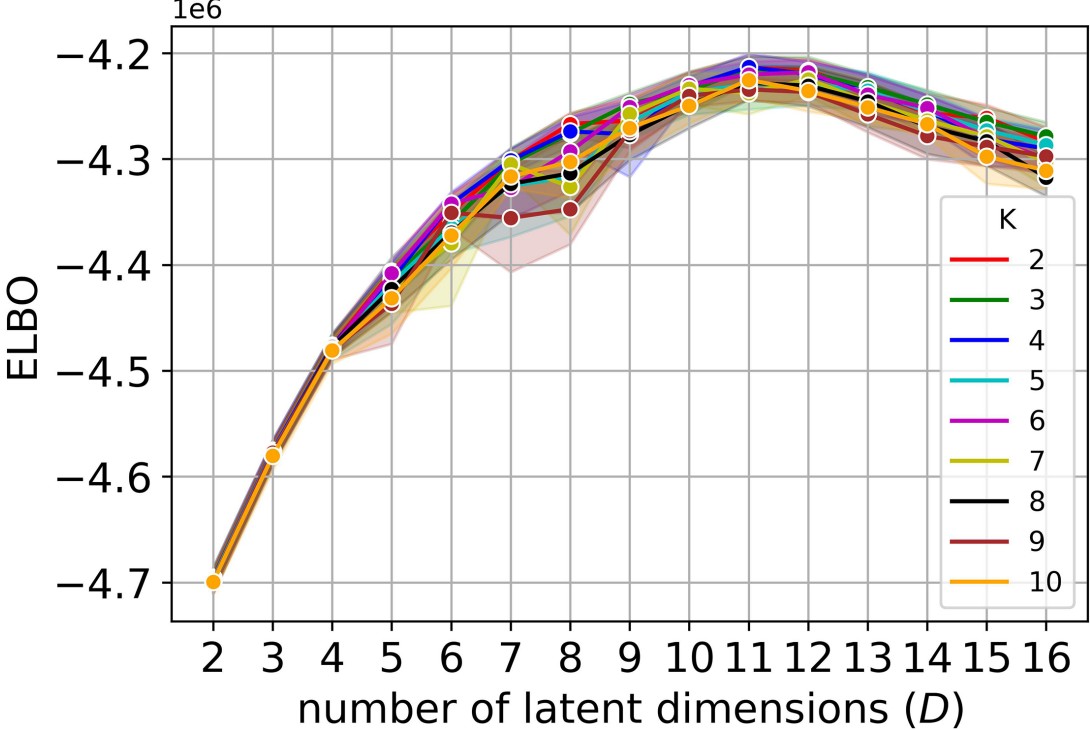

**Appendix 1—figure 1.** Model fit evidence lower bound (ELBO) criterion as a function of the number of latent dimensions determined for multiple values of the number of states (*K*). Error bars indicate the 95% confidence interval based on standard error across leave-one-out samples.

### Determining the number of states

Data from the exploratory dataset were also used to determine the number of states *K*. We investigated values from 2 to 10 and evaluated the consistency of estimated states across leave-one-participant-out folds. Specifically, we determined state sequences (one state per time step) for different *K* values. We followed the intuition that state sequences that were most consistent across data samples would be 'better' (*Ulrike von, 2010*). Consistency scores were calculated between all pairs of leave-one-participant-out samples using normalized mutual information. We chose the value *K* = 6 based on the figure below.

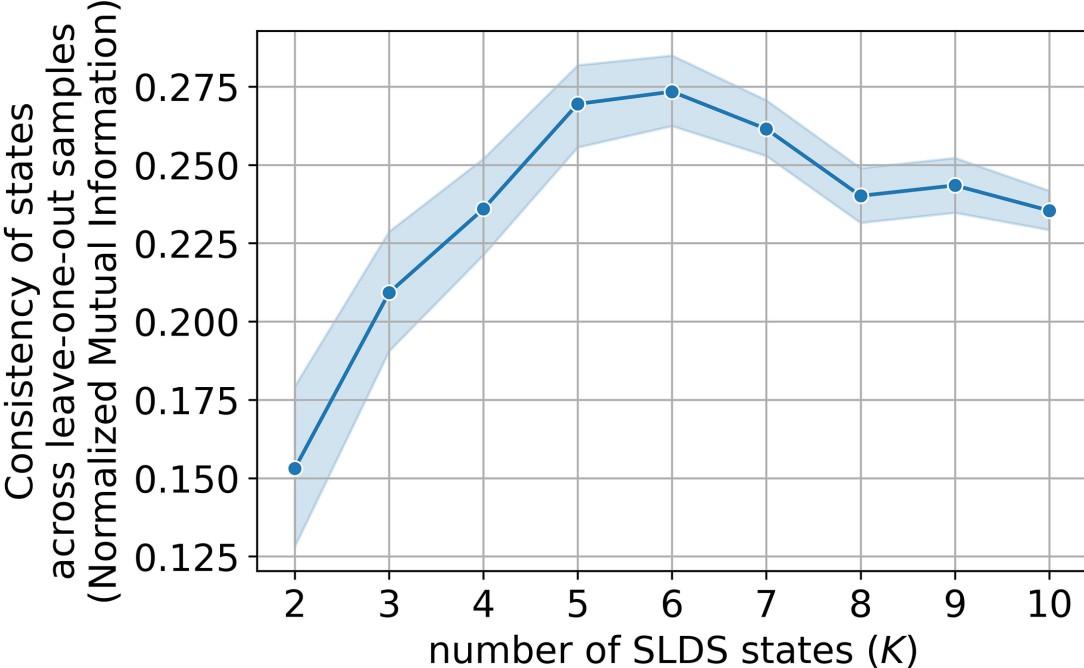

**Appendix 1—figure 2.** Consistency of state sequences across left-out participants as a function of the number of states ($K$). Error bars indicate the 95% confidence interval based on standard error across leave-one-out samples.

### Statistically significant state transitions

We determined state transitions that occurred with frequencies higher than what would be expected by chance. Recall that state transitions were determined based on the state transition probability matrix. Our procedure was as follows. Transition matrices were estimated for each bootstrap sample. To determine a null distribution of transition matrices, for every bootstrap sample, we temporally shuffled state sequences while preserving self-transitions (here, state transitions were the main object of study, not self-transitions). For each state, randomization of the state sequence was performed by shuffling contiguous blocks attributed to the same state (instead of shuffling at the time step level). Significant state transitions were those for which the mean of the transition probability (based on bootstrap samples) was greater than the mean based on the shuffled versions (paired Wilcoxon signed-rank test, $p < 0.05$). To correct for multiple comparisons ($K \times (K-1)$) transitions while ignoring self-transitions, we applied Bonferroni correction.

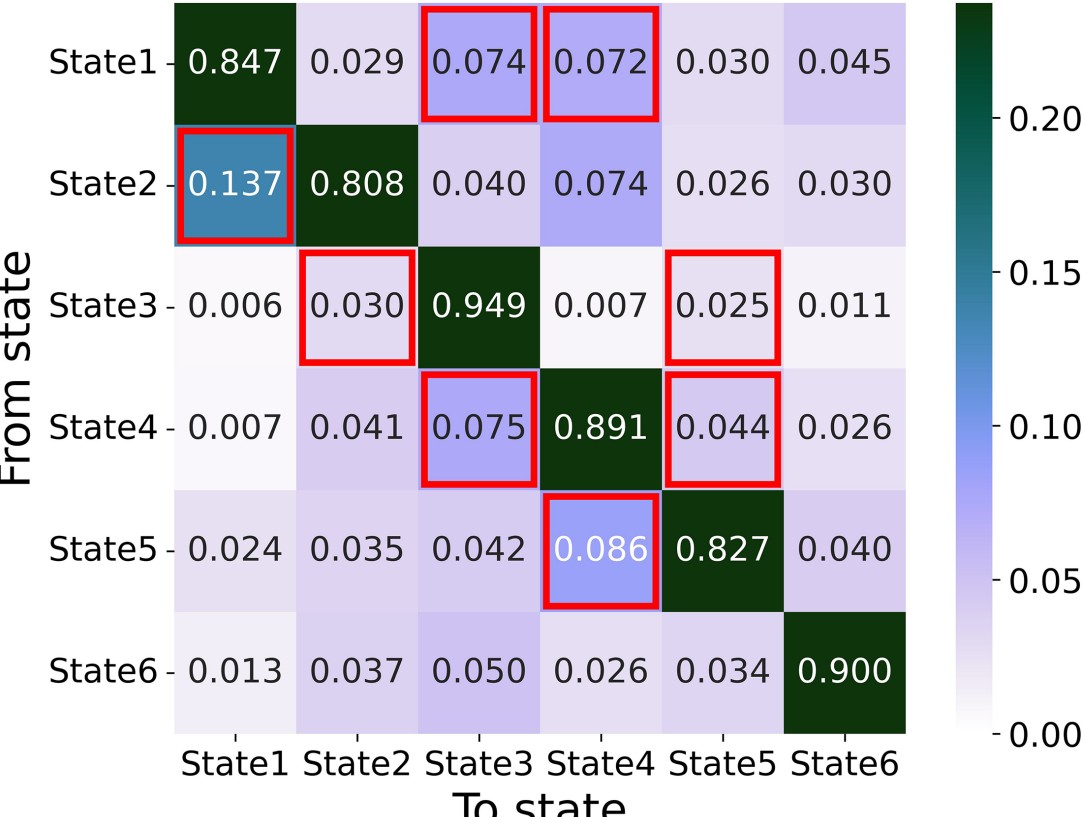

**Appendix 1—figure 3.** Average state transition matrix across bootstrap samples. State transitions highlighted in red have occurred more frequently than expected by chance ($p < 0.05$, corrected for multiple comparisons).

## Effect of the number of input stimulus categories (bins) in the analysis of *Figure 2*

We evaluated how the choice of input categories (bins) impacted the results shown in *Figure 2*. Overall, the results generalized well to 10 and 30 bins, instead of the original 20. However, we also note that a 'near-miss' state (STATE 5) was not well captured with only 10 bins given that it included an input ($A_5$) that included the circle collision, and it did not include the retreat input that was closest to the collision ($R_4$).

## Effect of the number of stimulus categories (bins) in the analysis of *Figure 7*

We also evaluated how the number of input stimulus categories (bins) impacted the results of the analysis shown in *Figure 7*. Broadly, the results generalized well across the number of bins used for inputs. However, some differences were observed. We discuss part (A) first. For example, in STATE 2 (shock/peri-shock) in the original case with 20 bins and with 10 bins, inputs at $R_9$ (20 bins) and $R_5$ (10 bins) steered the system to STATE 1 (post-shock); in contrast, such post-shock influence of the inputs was not detected with 30 bins. In addition, for STATE 1 (post-shock), inputs after the shock steered the trajectory away from the state (input $R_8$ with 20 bins and $R_{12}$ with 30 bins); in contrast, no corresponding tendency was observed with 10 bins. We discuss part (B) next. Input contributions to state transitions were consistent across the binning of inputs with one exception: no input was found to significantly steer the trajectory away from STATE 3 in the case of the STATE 3 ↦ STATE 5 transition when 10 bins were used (see second row).

