## [Editor Report · eLife Assessment]

Using highly sophisticated switching linear dynamical systems (SLDS) analyses applied to functional MRI data, this study provides **important** insights into network dynamics underlying threat processing. After identifying distinct neural network states associated with varying levels of threat proximity, the paper provides **compelling** evidence of intrinsically and extrinsically driven contributions to these within-state dynamics and between-state transitions. Although the findings could be made more biologically meaningful, this work will be of interest to a wider functional neuroimaging and systems neuroscience community.

---

## [Referee Report · Reviewer #1 (Public review)]

Summary:

The manuscript uses state-of-the-art analysis technology to document the spatio-temporal dynamics of brain activity during the processing of threats. The authors offer convincing evidence that complex spatio-temporal aspects of brain dynamics are essential to describe brain operations during threat processing.

Strengths:

Rigorous complex analyses well suited to the data.

Weaknesses:

Lack of a simple take-home message about discovery of a new brain operation.

Comments on revisions:

The authors have improved the presentation of the work. The overstatements about existing models of brain functions ignoring exogenous components remains largely unaddressed. The clarifications and improvements provided in revision confirm my assessment of the paper.

---

## [Referee Report · Reviewer #2 (Public review)]

Summary:

This paper by Misra and Pessoa uses switching linear dynamical systems (SLDS) to investigate the neural network dynamics underlying threat processing at varying levels of proximity. Using an existing dataset from a threat-of-shock paradigm in which threat proximity is manipulated in a continuous fashion, the authors first show that they can identify states that each have their own linear dynamical system and are consistently associated with distinct phases of the threat-of-shock task (e.g., "peri-shock", "not near", etc). They then show how activity maps associated with these states are in agreement with existing literature on neural mechanisms of threat processing, and how activity in underlying brain regions alters around state transitions. The central novelty of the paper lies in its analyses of how intrinsic and extrinsic factors contribute to within-state trajectories and between-state transitions. Additional analyses furthermore show how individual brain regions contribute to state dynamics. Finally, the authors show how their findings generalize to another (related) threat paradigm.

Strengths:

The analyses for this study are conducted at a very high level of mathematical and theoretical sophistication. The paper is very well written and effectively communicates complex concepts from dynamical systems. The paper provides valuable neuroscientific insights into threat processing, and the methodology has potential to deepen our understanding at a neurobiological level in future work.

Weaknesses:

I was somewhat disappointed initially by the level of inferences made by the authors based on their analyses at the level of systems neuroscience. After revision this has improved, for instance with inclusion of analyses on the importance of individual brain regions to state dynamics, but I still believe the findings can be made more biologically meaningful, for instance by focusing on what we learn from these sophisticated analyses beyond what is already known from more conventional methodologies. However, the paper as it stands is solid scientific work and such efforts may also be left to future work.

---

## [Author Response]

The following is the authors’ response to the original reviews

**Reviewer 1 (Public Review):**
SummaryThe manuscript uses state-of-the-art analysis technology to document the spatio-temporal dynamics of brain activity during the processing of threats. The authors offer convincing evidence that complex spatio-temporal aspects of brain dynamics are essential to describe brain operations during threat processing.StrengthsRigorous complex analyses well suited to the data.WeaknessesLack of a simple take-home message about discovery of a new brain operation.

We have addressed the concern under response to item 1 in Recommendations for the authors of Reviewer 2 below.

**Reviewer 1 (Recommendations for the authors):**
The paper presents sophisticated analyses of how the spatiotemporal activity of the brain processes threats. While the study is elegant and relevant to the threat processing literature, it could be improved by better clarification of novelty, scope, assumptions and implications. Suggestions are reported below.(1) Introduction: It is difficult to understand what is unsatisfactory in the present literature and why we need this study. For example, lines 57-64 report what works well in the work of Anderson and Fincham but do not really describe what this approach lacks, either in failing to explain real data in conceptual terms.

We have edited the corresponding lines to better describe what such approaches generally lack:

Introduction; Lines 63-66: However, the mapping between brain signals and putative mental states (e.g., “encoding”) remained speculative. More generally, state-based modeling of fMRI data would benefit from evaluation in contexts where the experimental paradigm affords a clearer mapping between discovered states and experimental manipulation.

(2) Also, based on the introduction it is unclear if the focus is on understanding the processing of threat or in the methodological development of experimental design and analysis paradigms for more ecologically valid situations.

In our present work, we tried to focus on understanding dynamics of threat processing while also contributing to methodological development of analysis of dynamic/ecologically inspired experiments. To that end, we have added a new paragraph at the end of Introduction to clarify the principal focus of our work:

Introduction; Lines 111-118: Is the present contribution focused on threat processing or methodological developments for the analysis of more continuous/ecologically valid paradigms? Our answer is “both”. One goal was to contribute to the development of a framework that considers brain processing to be inherently dynamic and multivariate. In particular, our goal was to provide the formal basis for conceptualizing threat processing as a dynamic process (see (Fanselow and Lester, 1987)) subject to endogenous and exogenous contributions. At the same time, our study revealed how regions studied individually in the past (e.g., anterior insula, cingulate cortex) contribute to brain states with multi-region dynamics.

(3) The repeated statement, based on the Fiete paper, that most analyses or models of brain activity do not include an exogenous drive seems an overstatement. There is plenty of literature that not only includes exogenous drives but also studies and documents them in detail. There are many examples, but a prominent one is the study of auditory processing. Essentially all human brain areas related to hearing (not only the activity of individual areas but also their communication) are entrained by the exogenous drive of speech (e.g. J. Gross et al, PLoS Biology 11 e1001752, 2013).

We have altered the original phrasing, which now reads as:

Introduction; Lines 93-95: Importantly, we estimated both endogenous and exogenous components of the dynamics, whereas some past work has not modeled both contributions (see discussion in (Khona and Fiete, 2022)).

Discussion; Lines 454-455: Work on dynamics of neural circuits in systems neuroscience at times assumes that the target circuit is driven only by endogenous processes (Khona and Fiete, 2022).

(4) Attractor dynamics is used as a prominent descriptor of fMRI activity, yet the discussion of how this may emerge from the interaction between areas is limited. Is it related to the way attractors emerge from physical systems or neural networks (e.g. Hopfield?).

This is an important question that we believe will benefit from computational and mathematical modeling, but we consider it beyond the scope of the present paper.

(5) Fig 4 shows activity of 4 regions, not 2 s stated in lines 201-202. Correct?

Fig. 4 shows activity of two regions and also the average activity of regions belonging to two resting-state networks engaged during threat processing (discussed shortly after lines 201-202). To clarify the above concern, we have changed the following line:

Results; Lines 228-230: In Fig. 4, we probed the average signals from two resting-state networks engaged during threat-related processing, the salience network which is particularly engaged during higher threat, and the default network which is engaged during conditions of relative safety.

(6) It would be useful to state more clearly how Fig 7B, C differs from Fig 2A, B (my understanding it is that in the former they are isolating the stimulus-driven processes)

We have clarified this by adding the following line in the Results:

Results; Lines 290-292: Note that in Fig. 7B/C we evaluated exogenous contributions only for stimuli associated with each state/state transition reported in Fig. 2A/B (see also Methods).

**Reviewer 2 (Public Review):**
SummaryThis paper by Misra and Pessoa uses switching linear dynamical systems (SLDS) to investigate the neural network dynamics underlying threat processing at varying levels of proximity. Using an existing dataset from a threat-of-shock paradigm in which threat proximity is manipulated in a continuous fashion, the authors first show that they can identify states that each has their own linear dynamical system and are consistently associated with distinct phases of the threat-of-shock task (e.g., “peri-shock”, “not near”, etc). They then show how activity maps associated with these states are in agreement with existing literature on neural mechanisms of threat processing, and how activity in underlying brain regions alters around state transitions. The central novelty of the paper lies in its analyses of how intrinsic and extrinsic factors contribute to within-state trajectories and betweenstate transitions. A final set of analyses shows how the findings generalize to another (related) threat paradigm.StrengthsThe analyses for this study are conducted at a very high level of mathematical and theoretical sophistication. The paper is very well written and effectively communicates complex concepts from dynamical systems. I am enthusiastic about this paper, but I think the authors have not yet exploited the full potential of their analyses in making this work meaningful toward increasing our neuroscientific understanding of threat processing, as explained below.Weaknesses(1) I appreciate the sophistication of the analyses applied and/or developed by the authors. These methods have many potential use cases for investigating the network dynamics underlying various cognitive and affective processes. However, I am somewhat disappointed by the level of inferences made by the authors based on these analyses at the level of systems neuroscience. As an illustration consider the following citations from the abstract: “The results revealed that threat processing benefits from being viewed in terms of dynamic multivariate patterns whose trajectories are a combination of intrinsic and extrinsic factors that jointly determine how the brain temporally evolves during dynamic threat” and “We propose that viewing threat processing through the lens of dynamical systems offers important avenues to uncover properties of the dynamics of threat that are not unveiled with standard experimental designs and analyses”. I can agree to the claim that we may be able to better describe the intrinsic and extrinsic dynamics of threat processing using this method, but what is now the contribution that this makes toward understanding these processes?

We have addressed the concern under response to item 1 in Recommendations for the authors below.

(2) How sure can we be that it is possible to separate extrinsically and intrinsically driven dynamics?

We have addressed the concern under response to item 2 in Recommendations for the authors below.

**Reviewer 2 (Recommendations for the authors):**
(1) To address the first point under weaknesses above: I would challenge the authors to make their results more biologically/neuroscientifically meaningful, in particular in the sections (in results and/or discussion) on how intrinsic and extrinsic factors contribute to within-state trajectories and between-state transitions, and make those explicit in both the abstract and the discussion (what exactly are the properties of the dynamics of threat that are uncovered?). The authors may also argue that the current approach lies the groundwork for such efforts, but does not currently provide such insights. If they would take this position, that should be made explicit throughout (which would make it more of a methodological paper).

The SLDS approach provides, we believe, a powerful framework to describe system-level dynamics (of threat processing in the the present case). A complementary type of information can be obtained by studying the contribution of individual components (brain regions) within the larger system (brain), an approach that helps connect our approach to studies that typically focus on the contributions of individual regions, and contributes to providing more neurobiological interpretability to the results. Accordingly, we developed a new measure of region importance that captured the extent to which individual brain regions contributed to driving system dynamics during a given state.

Abstract; Lines 22-25: Furthermore, we developed a measure of region importance that quantifies the contributions of an individual brain region to system dynamics, which complements the system-level characterization that is obtained with the state-space SLDS formalism.

Introduction; Lines 95-99: A considerable challenge in state-based modeling, including SLDS, is linking estimated states and dynamics to interpretable processes. Here, we developed a measure of region importance that provides a biologically meaningful way to bridge this gap, as it quantifies how individual brain regions contribute to steering state trajectories.

Results; Lines 302-321: Region importance and steering of dynamics: Based on time series data and input information, the SLDS approach identifies a set of states and their dynamics. While these states are determined in the latent space, they can be readily mapped back to the brain, allowing for the characterization of spatiotemporal properties across the entire brain. Since not all regions contribute equally to state properties, we propose that a region’s impact on state dynamics serves as a measure of its importance.

We illustrate the concept for STATE 5 (“near miss”) in Fig. 8 (see Fig. S17 for all states). Fig. 8A shows importance in the top row and activity below as a function of time from state entry.The dynamics of importance and activity can be further visualized (Fig. 8B), where some regions of particularly high importance are illustrated together with the ventromedial PFC, a region that is typically not engaged during high-threat conditions. Notably, the importance of the dorsal anterior insula increased quickly in the first time points, and later decreased. In contrast, the importance of the periaqueductal gray was relatively high from the beginning of the state and decreased moderately later.

Fig. 8C depicts the correlation between these measures as a function of time. For all but STATE 1, the correlation increased over time. Interestingly, for STATES 4-5, the correlation was low at the first and second time points of the state (and for STATE 2 at the first time point), and for STATE 3 the measures were actually anticorrelated; both cases indicate a dissociation between activity and importance. In summary, our results illustrate that univariate region activity can differ from multivariate importance, providing a fruitful path to understand how individual brain regions contribute to collective dynamic properties.

Discussion; Lines 466-487: In the Introduction, we motivated our study in terms of determining multivariate and distributed patterns of activity with shared dynamics. At one end of the spectrum, it is possible to conceptualize the whole brain as dynamically evolving during a state; at the other end, we could focus on just a few “key” regions, or possibly a single one (at which point the description would be univariate). Here, we addressed this gap by studying the importance of regions to state dynamics: To what extent does a region steer the trajectory of the system? From a mathematical standpoint, our proposed measure is not merely a function of activity of a region but also of the coefficients of the dynamics matrix capturing its effect on across-region dynamics (Eichler, 2005; Smith et al., 2010).

How distributed should the dynamics of threat be considered? One answer to this question is to consider the distribution of importance values for all states. For STATE 1 (“post shock”), a few regions displayed the highest importance values for a few time points. However, for the other states the distribution of importance values tended to be more uniform at each time point. Thus, based on our proposed importance measure, we conclude that threat-related processing is profitably viewed as substantially distributed. Furthermore, we found that while activity and importance were relatively correlated, they could also diverge substantially. Together, we believe that the proposed importance measure provides a valuable tool for understanding the rich dynamics of threat processing. For example, we discovered that the dorsal anterior insula is important not only during high-anxiety states (such as STATE 5; “near miss”) but also, surprisingly, for a state that followed the aversive shock event (STATE 1; “post shock”). Additionally, we noted that posterior cingulate cortex, widely known to play a central role in the default mode network, to have the highest importance among all other regions in driving dynamics of low-anxiety states (such as STATE 3 and STATE 4; “not near”).

Methods; Lines 840-866: Region importance We performed a “lesion study”, where we quantified how brain regions contribute to state dynamics by eliminating (zeroing) model parameters corresponding to a given region, and observing the resulting changes in system dynamics. According to our approach, the most important regions are those that cause the greatest change in system dynamics when eliminated.

The SLDS model represents dynamics in a low dimensional latent space and model parameters are not readily available at the level of individual regions. Thus, the first step was to project the dynamics equation onto the brain data prior to computing importance values. Thus, the linear dynamics equation in the latent space (Eq. 2) was mapped to the original data space of N = 85 ROIs using the emissions model (Eq. 1):\begin{document}$$\displaystyle  Y_{t} & =C X_{t}+d \\ & =C\left(A_{Z_{t}} X_{t-1}+V_{Z_{t}} U_{t}+b_{Z_{t}}\right)+d \\ & =C A_{Z_{t}} C^{\dagger} C X_{t-1}+C V_{Z_{t}} U_{t}+C b_{Z_{t}}+d \\ & =C A_{Z_{t}} C^{\dagger}\left(C X_{t-1}+d-d\right)+C V_{Z_{t}} U_{t}+C b_{Z_{t}}+d \\ & =\underbrace{\left(C A_{Z_{t}} C^{\dagger}\right)}_{\equiv \overparen{A_{Z_{t}}}} Y_{t-1}+\underbrace{\left(C V_{Z_{t}}\right)}_{\equiv \overparen{V_{Z_{t}}}} U_{t}+\underbrace{\left(C b_{Z_{t}}+\left(I-C A_{z_{t}} C^{\dagger}\right) d\right)}_{\equiv \widetilde{Z}_{t}} \\ & =\widehat{A_{Z_{t}}} Y_{t-1}+\widehat{V_{Z_{t}}} U_{t}+\widehat{b_{Z_{t}}},$$\end{document}

where *C*^†^ represents the Moore-Penrose pseudoinverse of *C*, and \begin{document}$\widehat{A_{Z_{t}}} \in \mathbb{R}^{N \times N}, \widehat{V_{Z_{t}}} \in \mathbb{R}^{N \times M}$\end{document} and \begin{document}$\widehat{b_{Z_{t}}} \in \mathbb{R}^{N}$\end{document} denote the corresponding dynamics matrix, input matrix, and bias terms in the original data space.

Based on the above, we defined the importance of the *i*^th^ ROI at time t based on quantifying the impact of “lesioning” the *i*^th^ ROI, i.e., by setting the *i*^th^ column of \begin{document}$\widehat{A_{Z_{t}}}$\end{document}, the *i*^th^ row of \begin{document}$\widehat{V_{Z_{t}}}$\end{document}, and the *i*^th^ element of \begin{document}$\widehat{b_{Z_{t}}}$\end{document} to 0, denoted \begin{document}${\widehat{A_{Z_{1}}}}^{(-i)}$\end{document}, *i*^th^ ROI \begin{document}${\widehat{V_{Z_{t}}}}^{(-i)}$\end{document}, and \begin{document}${\widehat{b_{Z_{t}}}}^{(-i)}$\end{document} respectively. Formally, the importance of the \begin{document}$q_{t}^{i}$\end{document} was defined as:\begin{document}$$\displaystyle  q_t^{i} & =  \left\lVert \left( \widehat{A}_{Z_t} Y_{t-1} + \widehat{V}_{Z_t} U_t + \widehat{b}_{Z_t} \right) - \left( \widehat{A}_{Z_t}^{(-i)} Y_{t-1} + \widehat{V}_{Z_t}^{(-i)} U_t + \widehat{b}_{Z_t}^{(-i)} \right) \right\rVert_2 \\[1.2ex] & =  \left\lVert \left( \widehat{A}_{Z_t} Y_{t-1} - \widehat{A}_{Z_t}^{(-i)} Y_{t-1} \right) + \left( \widehat{V}_{Z_t} U_t - \widehat{V}_{Z_t}^{(-i)} U_t \right) + \left( \widehat{b}_{Z_t} - \widehat{b}_{Z_t}^{(-i)} \right) \right\rVert_2 \\[1.2ex] & =  \left\lVert Y_{t-1}^{i} * \widehat{A}_{Z_t}^{(i)} + \left( \widehat{V}_{Z_t}^{(i)} U_t \right) * \mathbb{1}_i + \widehat{b}_{Z_t}^{(i)} * \mathbb{1}_i \right\rVert_2, $$\end{document}

where ‘∗’ indicates element-wise multiplication of a scalar with a vector, \begin{document}$Y_{t-1}^{i}$\end{document} is the activity of *i*^th^ ROI at time \begin{document}$t-1,{\widehat{A Z_{t}}}^{(i)}$\end{document} corresponds to the *i*^th^ column of *i*
\begin{document}$\widehat{A_{Z_{t}}},\left({\widehat{V_{Z_{t}}}}^{(i) \top} U_{t}\right)$\end{document} is the inner product between ^th^ row of *i*\begin{document}$\widehat{V_{Z_{t}}}$\end{document} and input ^th^ element of \begin{document}$U_{t},{\widehat{b_{Z}}}^{(1)}$\end{document} corresponds to the *i*^th^ ROI. Note that the term is a function of both the *i*\begin{document}$\widehat{b_{Z_{t}}}$\end{document} and ^th^ ROI’s activity as well as the coefficients of the dynamics matrix capturing the effect of region *i* on the one-step dynamics of the entire system (Eichler, 2005; Smith et al., 2010); the remaining terms capture the effect of the external inputs and the bias term on the one-step dynamics of the \begin{document}$\mathbb{1}_{i} \in \mathbb{R}^{N}$\end{document} represents an indicator vector corresponding to the *i*^th^ ROI.

After computing \begin{document}$q_{t}^{i}$\end{document} for a given run, the resultant importance time series was normalized to zero mean and unit variance.

(2) To address the second point under the weaknesses above: Given that the distinction between intrinsic and extrinsic dynamics appears central to the novelty of the paper, I would suggest the authors explicitly address this issue in the introduction and/or discussion sections.

The distinction between intrinsic and extrinsic dynamics is a modeling assumption of SLDS. We used such an assumption because in experimental designs with experimenter manipulated inputs one can profitably investigate both types of contribution to dynamics. While we should not reify the model’s assumption, we can gain confidence in our separation of extrinsically and intrinsically driven dynamics through controlled experiments where we can manipulate external inputs, or by demonstrating time-scale separation of intrinsic and extrinsic dynamics and that they operate at different frequencies. This is an important question that requires additional computational/mathematical modeling, but we consider it beyond the scope of the current paper. We have added the following lines in the discussion section:

Discussion; Lines 521-528: A further issue that we wish to discuss is related to the distinction between intrinsic and extrinsic dynamics, which is explicitly modeled in our SLDS approach (see Methods, equation 2). We believe this is a powerful approach because in experimental designs with experimenter manipulated inputs, one can profitably investigate both types of contribution to dynamics. However, complete separation between intrinsic and extrinsic dynamics is challenging to ascertain. More generally, one can gain confidence in their separation through controlled experiments where external inputs are manipulated, or by demonstrating timescale separation of intrinsic and extrinsic dynamics.

(3) In the abstract, the statement “.. studies in systems neuroscience that frequently assume that systems are decoupled from external inputs” sounds paradoxical after first introducing how threat processing is almost exclusively studied using blocked and event-related task designs (which obviously rely on external inputs only). Please clarify this.

In this work, we wished to state that the SLDS framework characterizes both endogenous and exogenous contributions to dynamics, whereas some past work has not modeled both contributions. To clarify, we have changed the corresponding line:

Abstract; Lines 19-20: Importantly, we characterized both endogenous and exogenous contributions to dynamics.

(4) In the abstract, the first mention of circles comes out of the blue; the paradigm needs to be introduced first to make this understandable.

We have rephrased the corresponding text:

Abstract; Lines 14-17: First, we demonstrated that the SLDS model learned the regularities of the experimental paradigm, such that states and state transitions estimated from fMRI time series data from 85 regions of interest reflected threat proximity and threat approach vs. retreat.

(5) In Figure 3, the legend shows z-scores representing BOLD changes associated with states. However, the z-scores are extremely low (ranging between -.4 and .4). Can this be correct, given that maps are thresholded at _p < .*001 (i.e., z > 3*._09)? A similar small range of z-scores is shown in the legend of Fig 5. Please check the z-score ranges.

The p-value threshold used in Fig. 3 is based on the voxelwise t-test conducted between the participantbased bootstrapped maps and null maps (see Methods : State spatial maps : “To identify statistically significant voxels, we performed a paired t-test between the participant-based boostrapped maps and the null maps.”). Thus, the p-value threshold in the figure does not correspond to the z-scores of the groupaveraged state-activation maps. Similarly in Fig. 5, we only visualized the state-wise attractors on a brain surface map without any thresholding. The purpose of using a z-score color bar was to provide a scale comparable to that of BOLD activity.